# Failures of nerve regeneration caused by aging or chronic denervation are rescued by restoring Schwann cell c-Jun

**Laura J Wagstaff**[1†‡], **Jose A Gomez-Sanchez**[2†], **Shaline V Fazal**[1§], **Georg W Otto**[3], **Alastair M Kilpatrick**[4], **Kirolos Michael**[1], **Liam YN Wong**[1], **Ki H Ma**[5], **Mark Turmaine**[1], **John Svaren**[5], **Tessa Gordon**[6], **Peter Arthur-Farraj**[7], **Sergio Velasco-Aviles**[2,8], **Hugo Cabedo**[2,8], **Cristina Benito**[1], **Rhona Mirsky**[1], **Kristjan R Jessen**[1*]

[1]Department of Cell and Developmental Biology, University College London, London, United Kingdom; [2]Instituto de Neurociencias de Alicante, Universidad Miguel Hernández-CSIC, San Juan de Alicante, Spain; [3]University College London Great Ormond Street Institute of Child Health, London, United Kingdom; [4]Centre for Regenerative Medicine, Institute for Regeneration and Repair, University of Edinburgh, Edinburgh, United Kingdom; [5]Department of Comparative Biosciences, School of Veterinary Medicine, University of Wisconsin-Madison, Madison, United States; [6]Division of Plastic and Reconstructive Surgery, The Hospital for Sick Children, Toronto, Canada; [7]John Van Geest Centre for Brain repair, Department of Clinical Neurosciences, University of Cambridge, Cambridge, United Kingdom; [8]Hospital General Universitario de Alicante, ISABIAL, Alicante, Spain

*For correspondence:
k.jessen@ucl.ac.uk

†These authors contributed equally to this work

Present address: ‡Centre for Regenerative Medicine, Institute for Regeneration and Repair, The University of Edinburgh, Edinburgh BioQuarter,5 Little France Drive, Edinburgh, United Kingdom; §John Van Geest Centre for Brian Repair, Department of Clinical Neurosciences, University of Cambridge, Cambridge, United Kingdom

**Competing interests:** The authors declare that no competing interests exist.

**Abstract** After nerve injury, myelin and Remak Schwann cells reprogram to repair cells specialized for regeneration. Normally providing strong regenerative support, these cells fail in aging animals, and during chronic denervation that results from slow axon growth. This impairs axonal regeneration and causes significant clinical problems. In mice, we find that repair cells express reduced c-Jun protein as regenerative support provided by these cells declines during aging and chronic denervation. In both cases, genetically restoring Schwann cell c-Jun levels restores regeneration to control levels. We identify potential gene candidates mediating this effect and implicate Shh in the control of Schwann cell c-Jun levels. This establishes that a common mechanism, reduced c-Jun in Schwann cells, regulates success and failure of nerve repair both during aging and chronic denervation. This provides a molecular framework for addressing important clinical problems, suggesting molecular pathways that can be targeted to promote repair in the PNS.

## Introduction

Among mammalian systems, peripheral nerve is often hailed as a prime example of a tissue with a striking regenerative potential. Nerve injury triggers the reprograming of myelin and non-myelin (Remak) Schwann cells to adopt a repair Schwann cell phenotype specialized to support regeneration, and injured neurons activate a gene program that facilitates axon growth. Yet, paradoxically, the clinical outcome of nerve injuries remains poor, and nerve damage constitutes a significant clinical and economic burden. Remarkably, treatment of nerve injuries has not advanced significantly for decades (*Furey et al., 2007*; *Jonsson et al., 2013*; reviewed in *Fu and Gordon, 1995*; *Boyd and*

*Gordon, 2003a*; *Höke, 2006*; *Allodi et al., 2012*; *Scheib and Höke, 2013*; *Doron-Mandel et al., 2015*; *Jessen and Mirsky, 2016*; *Fawcett and Verhaagen, 2018*; *Jessen and Arthur-Farraj, 2019*).

The question of why a potentially regenerative tissue fails to respond effectively to injury and ensure clinical recovery is important both for promoting nerve repair, and also more generally. A number of other systems with experimentally established regenerative capacity, for example, skin, heart, and pancreatic islets, also fail to show clinically useful regenerative response to tissue damage (*Cohen and Melton, 2011*; *Eguizabal et al., 2013*; *Jessen et al., 2015*).

In the case of peripheral nerves, recent work has highlighted two important factors that prevent full expression of their regenerative potential. One is the age of the animal at the time of injury, increasing age resulting in a marked decrease in regeneration. The other is the adverse effect of chronic denervation on the nerve distal to injury, since this tissue gradually loses the capacity to support axon growth as it lies denervated during the often extensive time it takes regenerating axons to reach their targets. These two problems turn out to involve a common factor, namely a repair Schwann cell failure, since both during aging and chronic denervation, the denervated Schwann cells in the distal stump undergo molecular and morphological changes that result in a striking functional deterioration of these important drivers of axonal regeneration (reviewed in *Verdú et al., 2000*; *Sulaiman and Gordon, 2009*; *Painter, 2017*; *Jessen and Mirsky, 2019*).

In the present work, we have tested whether the dysfunction of repair Schwann cells in these two apparently unrelated situations relates to a common factor, namely a failure to activate or maintain high levels of the transcription factor c-Jun. That this might be so, is based on our previous finding that c-Jun, which is upregulated in Schwann cells in injured nerves, is a global amplifier of the repair Schwann cell phenotype (*Arthur-Farraj et al., 2012*; reviewed in *Jessen and Mirsky, 2016*; *Jessen and Mirsky, 2019*; *Jessen and Arthur-Farraj, 2019*), and on subsequent findings showing that enhanced Schwann cell c-Jun promotes regeneration, both through nerve grafts and in vitro (*Arthur-Farraj et al., 2012*; *Huang et al., 2015*; *Huang et al., 2019*).

The age-dependent decline in regenerative capacity of human and animal nerves is well established (*Pestronk et al., 1980*; *Tanaka and deF. Webster, 1991*; *Tanaka et al., 1992*; *Graciarena et al., 2014*; reviewed in *Vaughan, 1992*; *Verdú et al., 2000*; *Ruijs et al., 2005*). This is associated with a reduced initial inflammatory response followed by enhanced chronic inflammation (*Scheib and Höke, 2016*; *Büttner et al., 2018*). Interestingly, diminished regeneration is not caused by age-dependent changes in neurons. Rather, aging results in subdued activation of the repair Schwann cell phenotype after injury, including reduced c-Jun expression, resulting in regeneration failure (*Painter et al., 2014*; reviewed in *Painter, 2017*).

The other major barrier to repair that we consider here is caused by long-term denervation of nerves distal to injury. This is an important issue in human nerve regeneration (*Ruijs et al., 2005*) and has been studied in some detail in rats, revealing that chronic denervation results in reduced expression of repair-associated genes including *Gdnf*, *Bdnf*, *Ntf3*, and *Ngfr*, accompanied by a dramatic reduction in the ability of denervated distal stumps to support regeneration even of freshly transected axons (*Fu and Gordon, 1995*; *You et al., 1997*; *Sulaiman and Gordon, 2000*; *Höke et al., 2002*; *Michalski et al., 2008*; *Eggers et al., 2010*). There is direct evidence for a comparable deterioration of repair cells and repair capacity during chronic denervation of human nerves (*Wilcox et al., 2020*; reviewed in *Ruijs et al., 2005*). Chronic denervation also results in reduced repair cell numbers and shortening of repair cells (*Benito et al., 2017*; *Gomez-Sanchez et al., 2017*; reviewed in *Jessen and Mirsky, 2019*). Thus, the repair phenotype is not stable but fades with time after injury, thereby contributing to the poor outcome after nerve damage in humans.

Schwann cell reprograming after nerve injury involves upregulation of trophic factors and cytokines, activation of EMT genes, and myelin autophagy for myelin clearance and downregulation of myelin genes (*Brushart et al., 2013*; *Arthur-Farraj et al., 2017*; *Clements et al., 2017*; reviewed in *Gröthe et al., 2006*; *Chen et al., 2007*; *Gambarotta et al., 2013*; *Glenn and Talbot, 2013*; *Jessen and Mirsky, 2016*; *Boerboom et al., 2017*; *Jessen and Arthur-Farraj, 2019*; *Nocera and Jacob, 2020*). Myelin and Remak Schwann cells also increase in length by two-to-three fold and often branch as they convert to repair cells and form regeneration tracks, Bungner bands, that guide regenerating axons (*Gomez-Sanchez et al., 2017*). The molecular

signals involved in the decline of these repair-supportive features during aging and chronic denervation have not been known.

The transcription factor c-Jun regulates the reprograming of myelin and Remak cells to repair cells by accelerating the extinction of myelin genes, promoting myelin breakdown, and by amplifying the upregulation of a broad spectrum of repair-supportive features, including the expression of trophic factors. Accordingly, genetic removal of c-Jun from Schwann cells results in functionally impaired repair cells and regeneration failure (*Arthur-Farraj et al., 2012*; *Fontana et al., 2012*; reviewed in *Jessen and Arthur-Farraj, 2019*).

Here, we provide evidence that a common molecular mechanism, the dysregulation of c-Jun in Schwann cells, is central to two major categories of regeneration failure in the PNS. The high levels of Schwann cell c-Jun triggered by nerve injury in young animals are not achieved in older ones, and, irrespective of age, the elevated c-Jun expression seen after injury steadily decreases during long-term denervation. Importantly, we show that in both models of regeneration failure, genetically restoring Schwann cell c-Jun levels in vivo also restores regeneration rates to that in controls. By establishing c-Jun as an important regulator of the success and failure of nerve repair during aging and chronic denervation this observation provides a common molecular framework for addressing an important clinical problem, and suggests molecular pathways that can be targeted to promote repair in the PNS.

## Results

### In aging animals, maintaining c-Jun levels in Schwann cells reverses age-related defects in nerve regeneration

Age-dependent failure of nerve regeneration is accompanied by subdued elevation of c-Jun, a major regulator of the repair cell phenotype (*Painter et al., 2014*). To test whether this reduction in c-Jun controls the reduced capacity of these cells to support axon growth, we first compared c-Jun upregulation in young and older WT mice (*Figure 1A*). Four days after transection, c-Jun protein levels in the distal nerve stump in aged mice (8–10 months) were found to be ~50% lower than in young (6–8 weeks) mice.

To determine the functional significance of this, we studied $Mpz^{Cre+}$;$R26c$-$Junstop^{ff/+}$ mice (referred to as c-Jun OE/+ mice), which we generated previously (*Fazal et al., 2017*). In these mice, c-Jun levels are enhanced in Schwann cells only. In western blots of uninjured adult sciatic nerves of c-Jun OE/+ mice, c-Jun is elevated about seven fold compared to WT. While there is a modest reduction in myelin thickness, nerve architecture and Schwann cell morphology are normal (*Fazal et al., 2017*). We found that in c-Jun OE/+ mice, the age-dependent decline in c-Jun protein levels after sciatic nerve cut was prevented, and that c-Jun levels in the distal stump of young WT and aging c-Jun OE/+ mice were similar by western blots 4 days after cut (*Figure 1A*). At the mRNA level, a non-significant trend towards lower c-Jun expression was seen in 3-day cut nerves of aged WT nerves, while there was a significant elevation of c-Jun mRNA in cut c-Jun OE/+ nerves as expected (*Figure 1B*). c-Jun levels seen in western blots were confirmed in immunofluorescence experiments on 3-day cut nerves, using Sox10 antibodies to selectively identify Schwann cell nuclei, and c-Jun antibodies (*Figure 1C*). In WT mice, older nerves contained fewer c-Jun-positive Schwann cell nuclei and the labeling of the c-Jun-positive nuclei was weaker, compared to young nerves. In aged c-Jun OE/+ nerves nuclear c-Jun was restored to levels similar to those in young WT nerves.

Regeneration in young and aged WT mice and aged c-Jun OE/+ mice was compared using neuron back-filling, a method that provides an optimal measure of regenerative capacity by determining the number of neuronal cell bodies have regenerated axons through a nerve at a measured distance distal to injury (*Figure 1—figure supplement 1*; *Novikova et al., 1997*; *Boyd and Gordon, 2003b*; *Catapano et al., 2016*). Four days after sciatic nerve crush, retrograde tracer was applied to the distal stump 7 mm from the crush site. Seven days later, the animals were sacrificed and the number of back-filled DRG and spinal cord motor neurons were counted. The results were comparable for both neuronal populations (*Figure 1D,E*). The number of neurons regenerating through the distal stump of aged WT mice was reduced by about 50% compared to young mice. In aged c-Jun OE/+ mice on

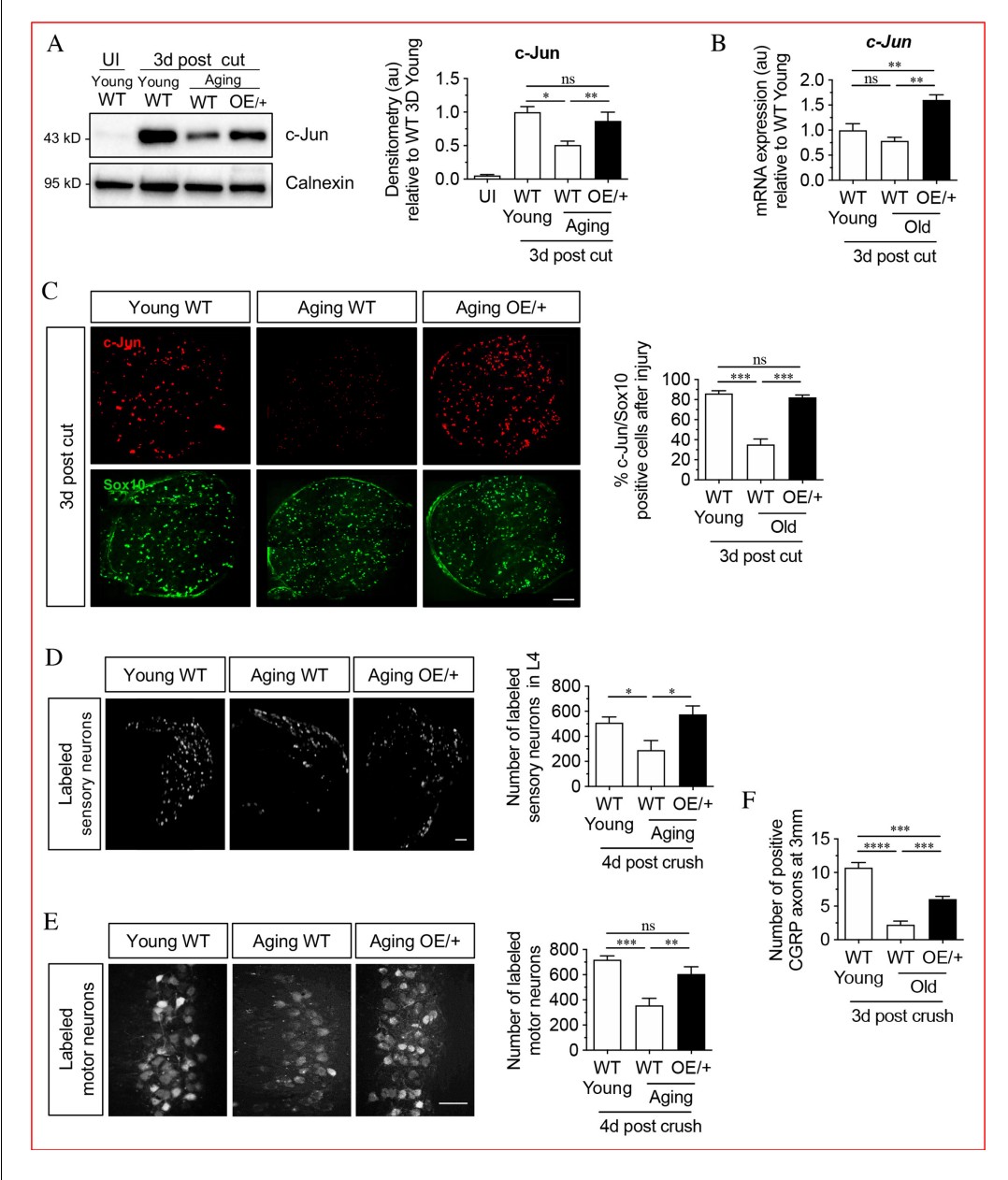

**Figure 1.** Restoring Schwann cell c-Jun protein reverses the age-related decline in nerve regeneration. (A) Representative western blots of c-Jun in young and aging WT nerves and aging c-Jun OE/+ nerves 3 days post-injury. The graph shows densitometric quantitation of the western blots. c-Jun upregulation is impaired in WT aged nerves but restored in aged c-Jun OE/+ nerves. Data are normalized to young WT 3 days post-cut; *p<0.05, **p<0.005, ns, non-significant. Young UI WT n = 6, n = 5 for all other experimental groups. (B) RTqPCR analysis of 3-day cut nerves. Data normalized to young WT 3 days post-cut.; **p<0.005, ns, non-significant, n = 4. (C) Representative images showing immunofluorescence of c-Jun in double labeling with Schwann cell nuclear marker Sox10 in sections of the distal nerve stump in young and aging WT and aging c-Jun OE/+ mice 3 days post-cut. In the graph, the results are quantitated by cell counting. In aging WT Schwann cells, c-Jun is reduced, but elevated to youthful levels in aging c-Jun OE/+ Schwann cells; **p<0.001, ns, non-significant. n = 3 for each experimental group. (D) Representative images showing Fluorogold-labeled sensory neurons in L4 DRGs of young WT and aging WT and c-Jun OE/+ mice 1-week post back-filling following a 4-day crush injury. The graph shows quantitation by cell counting. There is an age-related decrease in back-filled neurons in WT samples (p=0.0309), while the high number of regenerating neurons in young WT mice is maintained in aging c-Jun OE/+ DRGs (p=0.0211). Unpaired Student's t-test. Young WT n = 6, aging WT n = 5, aging c-Jun OE/+ n = 6. (E) Representative images of Fluorogold-labeled motor neurons in young WT and aging WT and c-Jun OE/+ mice 1-week post back-filling following a 4-day crush injury. The graph shows quantitation of the results. Counts of labeled motor neurons mirrors those of sensory neurons since in WT mice, but not in c-Jun OE/+ mice, the number of back-labeled motor neurons decreases with age; ***p<0.001, **p<0.005, n = 6 for all experimental groups. (F) Counts of calcitonin gene-related peptide (CGRP)+ regenerating axons 3 mm from crush injury of the sciatic nerve of young

*Figure 1 continued on next page*

*Figure 1 continued*

and aged WT mice, and of aging c-Jun OE/+ mice; ****p<0.0001, ***p<0.001. Young WT n = 5, aging WT and c-Jun OE/+ n = 6. All numerical data are analyzed by one-way ANOVA with Tukey's multiple comparison test and represented as means ± SEM. All scale bars: 100 µm.

The online version of this article includes the following figure supplement(s) for figure 1:

**Figure supplement 1.** Schematic representation of neuron backfilling.

the other hand, regeneration of DRG and motor neurons was restored to levels similar to those in young WT nerves.

In confirmation, counts of CGRP+ fibers in the sciatic nerve were performed 3 days after nerve crush, 3 mm from the injury site (*Figure 1F*). The number of fibers were reduced in aged WT nerves compared to young ones, but increased in the aged c-Jun OE/+ nerves.

These experiments confirm that the failure of repair cell function in older animals is accompanied by failure to fully upregulate c-Jun in Schwann cells after injury (*Painter et al., 2014*). Importantly, restoration of c-Jun elevation selectively in Schwann cells to that seen in young animals is sufficient to restore nerve regeneration to youthful levels.

## A mouse model of distal stump deterioration

We established a model of chronic denervation in mice, since previous studies have been carried out in rats. Sciatic nerves were cut followed by deflection of the proximal stump to leave the distal stump un-innervated for 1 week (short-term denervation) or 10 weeks (chronic denervation). mRNA levels for genes associated with denervated Schwann cells (*S100b*, *Ngfr*, *Gdnf*, and *sonic hedgehog (Shh)*) declined substantially between 1 and 10 weeks of denervation (*Figure 2A*). Western blots of p75NTR (Ngfr) in the distal stump showed a rise to a maximum 1 week after injury and a decline thereafter to <50% of peak levels at 10 weeks (*Figure 2B*) in line with that seen in rat (*You et al., 1997*).

Regeneration through freshly cut and long-term denervated nerve stumps were compared by back-filling of spinal cord motor neurons. For this, acutely cut (immediate repair) or 10-week denervated tibial distal stumps were sutured to freshly cut peroneal nerves (*Figure 2—figure supplement 1*) Two weeks later, Fluorogold retrograde tracer was applied to the tibial stumps 4 mm distal to the suturing site. One week after the application of tracer, the number of retrogradely labeled motor neurons in the spinal cord was counted. Only about half as many spinal cord motor neurons projected into the 10-week denervated stumps compared to the acutely transected stumps (*Figure 2C*).

Failure of regeneration through 10-week denervated stumps was confirmed by immunohistochemistry and counting of neurofilament-positive axons in similar experiments. Ten-week denervated stumps contained many fewer regenerating neurofilament labeled axons than nerve stumps sutured immediately after transection (*Figure 2D*).

Counting back-filled neurons following immediate repair or repair 1 week after transection, revealed a similar number of regenerating neurons (*Figure 2E*). This shows that the capacity of the distal stump to support regeneration declines between 1 and 10 weeks of denervation.

These experiments established a baseline for studying the effects of prolonged denervation on the capacity of mouse repair Schwann cells to support neuronal regeneration.

## c-Jun is downregulated in chronically denervated Schwann cells

We determined whether decline in Schwann cell c-Jun expression was involved in regeneration failure caused by chronic denervation, as seen during aging. Measuring c-Jun protein levels in distal stumps showed strong elevation 3 days and 1 week after sciatic nerve cut followed by a decline to ~40% of 1 week levels at 10 weeks (*Figure 3A*). We verified that the c-Jun shown in these western blots represented c-Jun in Schwann cells, using mice with conditional c-Jun inactivation selectively in Schwann cells (*Arthur-Farraj et al., 2012*; *Figure 3B*). Further, 1- and 10-week denervated stumps were compared using double immunofluorescent labeling with c-Jun antibodies and Sox10 antibodies to selectively identify Schwann cell nuclei. Ten-week denervated stumps showed many more c-Jun negative, Sox10-positive nuclei, and the c-Jun labeling, where present, was also weaker (*Figure 3C*).

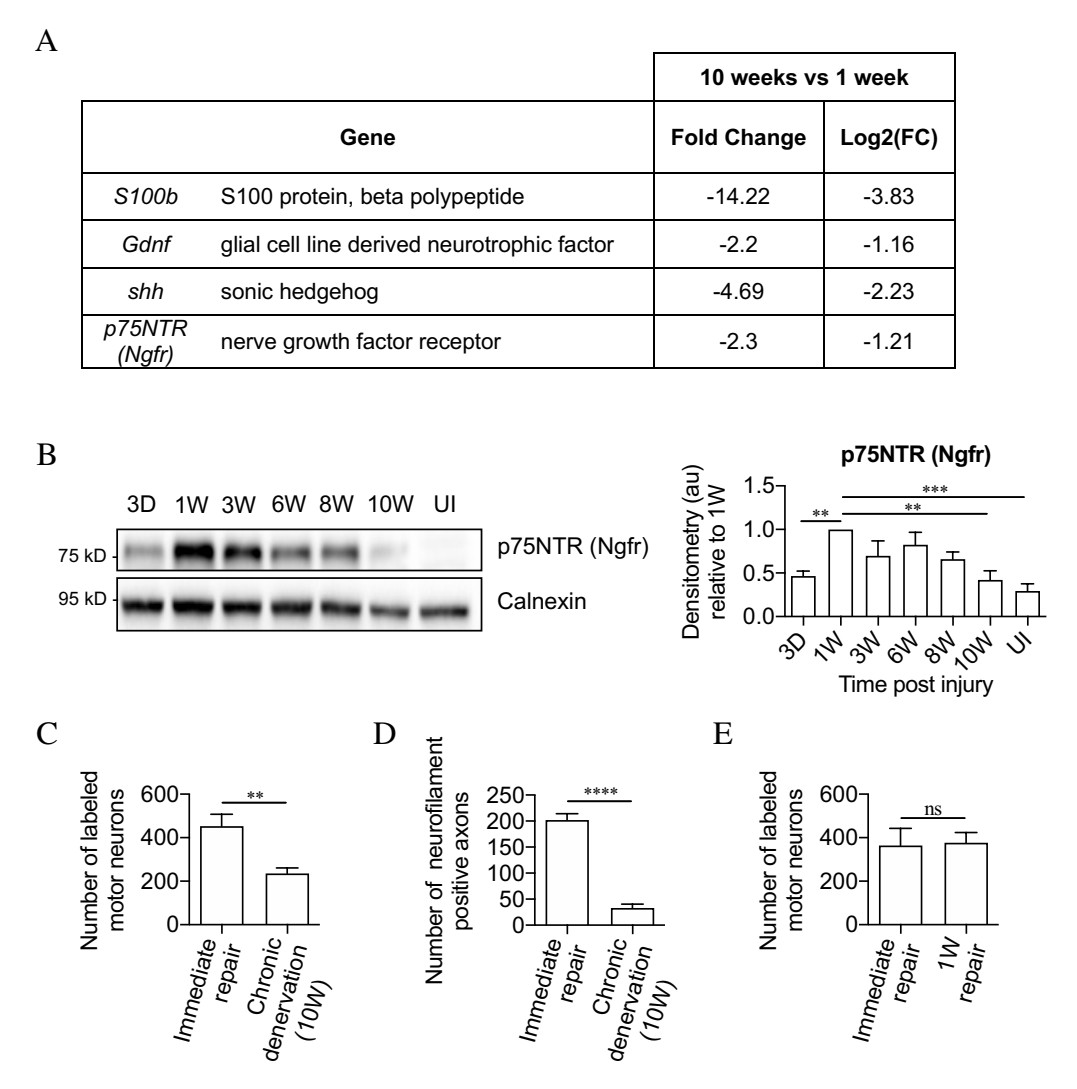

**Figure 2.** The mouse model of chronic denervation. (A) Analysis of RNA sequencing data showing decrease in gene expression during chronic denervation. (B) Representative western blot showing p75NTR expression in uninjured (UI) nerves and distal nerve stumps following 3 days, and 1, 3, 6, 8 and 10 weeks of denervation. The graph shows quantitation of the results. P75NTR peaks 1 week after injury and gradually declines during prolonged denervation. Data normalized to 1 week after injury. One-way ANOVA with Dunnett's multiple comparison test; **p<0.005, ***p<0.001. n = 4. (C) Counts of back-filled Fluorogold-labeled regenerating motor neurons following immediate repair or chronic 10-week denervation show a decrease in motor neuron regeneration into chronically denervated stumps. Unpaired Student's t-test; **p=0.0020. n = 6 for each time point. (D) Counts of neurofilament+ axons mirrors the decline in regeneration observed with chronic denervation shown in C. Counts were performed on transverse sections taken 3 mm from the repair site1 week after repair. Unpaired Student's t-test; ****p<0.0001. Immediate repair n = 5, chronic denervation n = 4. (E) Counts of back-filled Fluorogold-labeled motor neurons showing similar numbers of regenerating neurons following immediate repair or repair after 1 week of denervation. Unpaired Student's t-test; p=0.9. n = 3. All numerical data represented as means ± SEM.
The online version of this article includes the following figure supplement(s) for figure 2:

**Figure supplement 1.** Surgical procedures used to study regeneration after immediate repair and chronic denervation.

The decline in c-Jun and p75NTR expression after long-term denervation in vivo was mimicked in purified Schwann cells in vitro. By western blots, cells that had been maintained ~6 weeks in vitro (nine passages), contained less c-Jun and p75NTR protein compared to cells maintained ~10 days in vitro (two passages) (*Figure 3D*). The levels of c-Jun mRNA also declined in long-term cultures (*Figure 3E*). The reduction in c-Jun in Schwann cell nuclei was confirmed using double immunofluorescent labeling with c-Jun and Sox 10 antibodies to identify Schwann cells (*Figure 3F*). These in

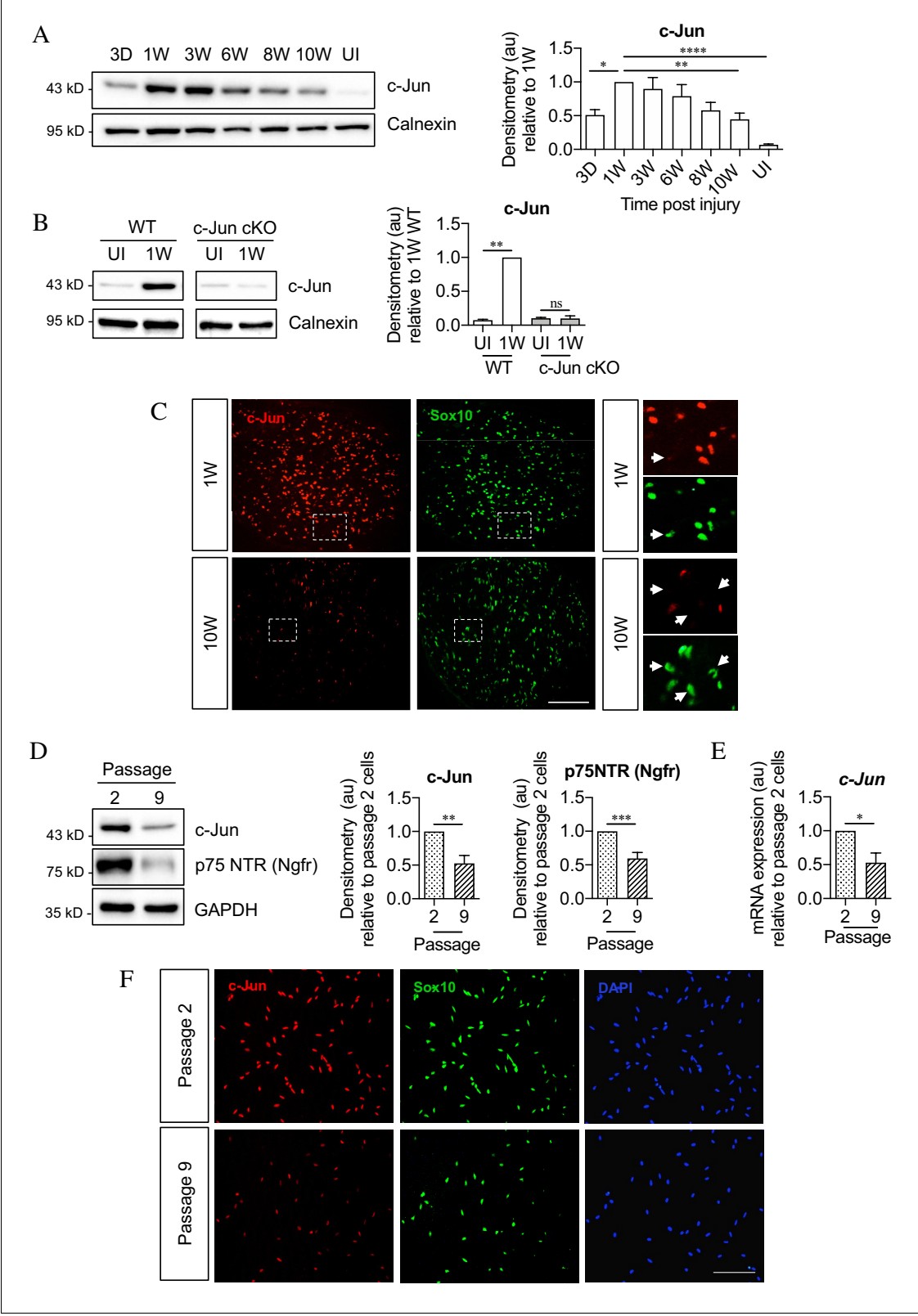

**Figure 3.** c-Jun declines in the distal nerve stump during chronic denervation and long-term culture. (**A**) Representative Western blot of c-Jun in WT uninjured (UI) nerves and distal stumps following 3 days and 1, 3, 6, 8, and 10 weeks of denervation. The graph quantitates the results, showing an initial increase followed by a decline in c-Jun levels. Data normalized to 1 week post-injury. One-way ANOVA with Dunnett's multiple comparisons test; *p<0.05, **p<0.005, ****p<0.0001. n = 5. (**B**) Representative western blot comparing c-Jun expression 1 week after injury in WT and c-Jun cKO mice.
*Figure 3 continued on next page*

*Figure 3 continued*

The graph quantitates the results, showing upregulation of c-Jun in WT nerves but not in c-Jun cKO nerves, demonstrating that the c-Jun upregulation after injury is Schwann cell specific. Data normalized to WT 1 week post-injury. Two-way ANOVA with Sidak's multiple comparison test; ****$p<0.0001$. n = 5. (C) Representative immunofluorescence images of c-Jun/Sox10 double labeling in transverse sections of the distal stumps 1 and 10 weeks after cut. Boxed areas shown at higher magnification in right hand panels. Note loss of c-Jun protein from Schwann cell nuclei at 10 weeks (arrows). (D) Representative Western blot of c-Jun and p75NTR in Schwann cell cultures after two or nine passages. The results are quantitated in the graphs, showing decline in c-Jun and p75NTR with time in vitro. Data normalized to passage 2. Unpaired Student's t-test; **$p=0.0023$, ***$p=0.0007$. n = 6 for c-Jun, n = 7 for p75NTR. (E) qPCR analysis showing reduction in c-Jun mRNA in Schwann cultures following nine passages. Data normalized to passage 2. Unpaired Student's t-test; *$p=0.0299$. n = 3. (F) c-Jun/Sox10 double labeling with nuclear marker DAPI after two or nine passages. Note decline of nuclear c-Jun in passage 9 cells. All numerical data represent means ± SEM.

vitro experiments suggest that the decline in c-Jun and p75NTR during chronic denervation is not driven by endoneurial signals.

## c-Jun downregulation during chronic denervation is prevented in c-Jun OE/+ mice

The decline in c-Jun levels at the same time as repair cells lose capacity to support regeneration raises the questions of whether the functional deterioration of these cells is partly a consequence of c-Jun reduction, and whether repair cells, and regeneration through chronically denervated nerves, would be maintained if c-Jun reduction was prevented.

We addressed this using the c-Jun OE/+ mice examined earlier in experiments on aging (previous section; *Fazal et al., 2017*). The mice were 6–8 weeks old at the time of injury, corresponding to young mice in the study on aging. One- and 3-week denervated distal stumps of c-Jun OE/+ and WT mice contained similar c-Jun protein levels. However, at 10 weeks, when c-Jun had declined in WT mice, c-Jun was maintained in c-Jun OE/+ mice at levels similar to those at 1 week (*Figure 4A*). This was confirmed by double label c-Jun/Sox10 immunofluorescence (*Figure 4B*). This showed similar c-Jun nuclear labeling in 1-week denervated stumps of WT and c-Jun OE/+ mice, while at 10 weeks, WT nerves showed a reduced number of c-Jun-positive nuclei and decreased labeling intensity. This decrease was prevented in c-Jun OE/+ nerves.

These experiments indicate that in 1- to 3-week cut nerves, the maximum capacity of Schwann cells to express c-Jun protein is already reached in the WT, both genotypes showing a similar 80–100 fold elevation after injury. During chronic denervation, these high expression levels fall substantially in WT nerves, but not in c-Jun OE/+ nerves. The c-Jun OE/+ mice can therefore be used to test whether the regeneration failure induced by chronic denervation is due to the failure to maintain high c-Jun protein levels.

## Maintaining c-Jun levels during chronic denervation prevents regeneration failure

Regeneration thorough chronically denervated distal stumps of WT and c-Jun OE/+ mice was compared using neuron back-filling. Acutely cut common peroneal nerves were sutured to acutely cut or 10-week denervated tibial nerves and allowed to regenerate for 2 weeks prior to application of Fluorogold retrograde tracer. In WT mice, the number of DRG neurons projecting through 10-week cut nerves was only about half that projecting through acutely cut nerves. In 10-week cut c-Jun OE/+ nerves, however, this drop was not seen (*Figure 5A*). Similar results were obtained for spinal cord motor neurons (*Figure 5B*). Thus, both DRG and motor neurons regenerated as well through 10-week denervated c-Jun OE/+ nerves as they did through acutely cut WT nerves, suggesting that c-Jun OE/+ Schwann cells maintain their capacity to support regenerating neurons despite chronic denervation. In confirmation, regeneration into 10-week denervated distal stumps of the tibial nerve was examined with neurofilament staining, 1 week after the stumps were sutured to freshly cut peroneal nerves. Nearly three times more fibers were found in 10-week denervated c-Jun OE/+ nerves compared to 10-week denervated WT nerves (*Figure 5C*). The number of fibers found in WT nerves after immediate repair is shown in *Figure 2D*.

We verified that the back-filling paradigm worked as expected in c-Jun OE/+ mice. First, peroneal nerves in WT and c-Jun OE/+ mice were transected followed by immediate application of tracer to the injured proximal stump. The number of back-filled DRG and motor neurons in c-Jun OE/+ mice

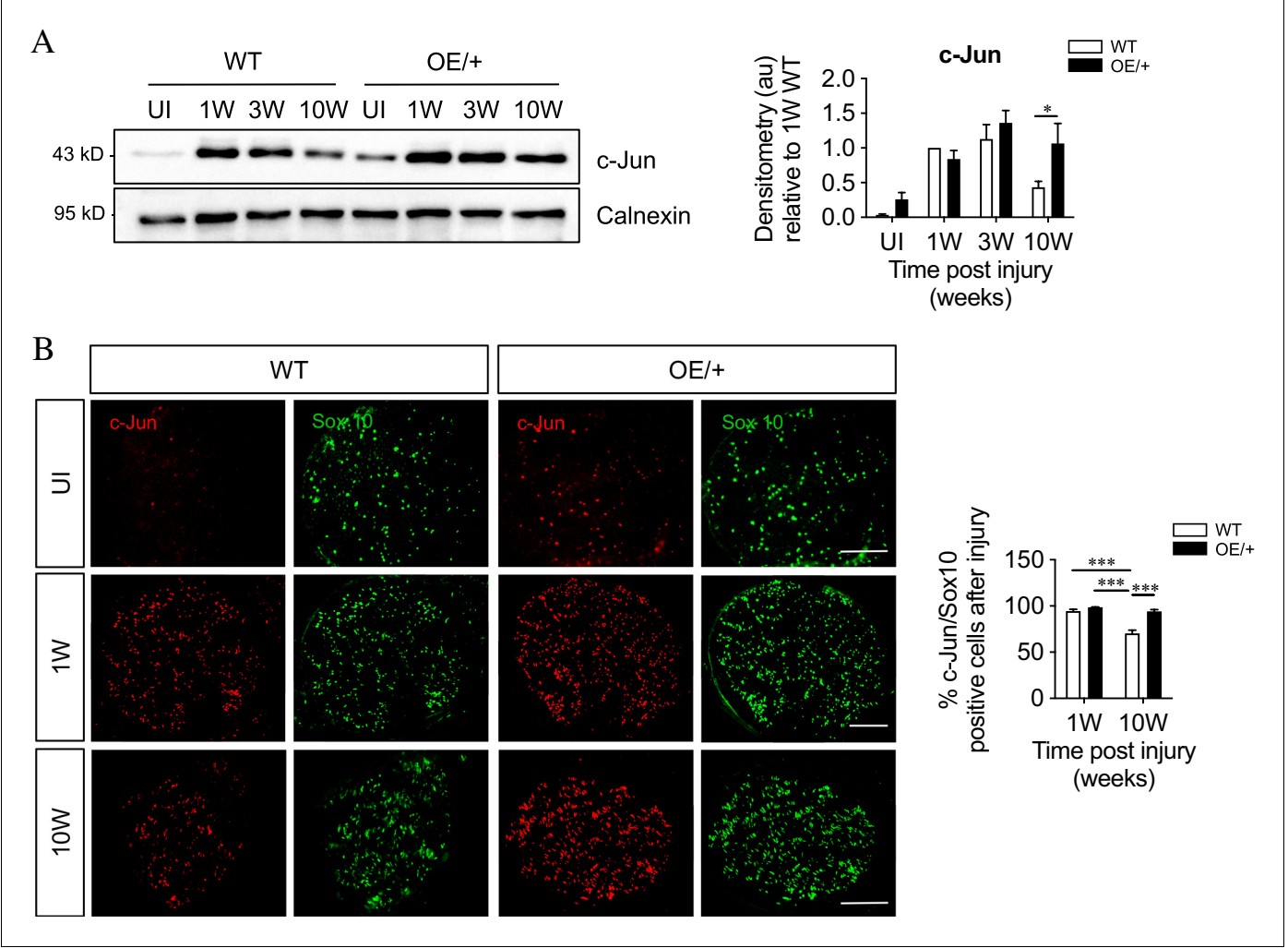

**Figure 4.** c-Jun expression is maintained in c-Jun OE/+ Schwann cells during chronic denervation. (**A**) Representative western blots of c-Jun in WT and c-Jun OE/+ distal stumps after 1, 3, and 10 weeks of denervation. The results are quantitated in the graph. In contrast to WT nerves, c-Jun OE/+ nerves maintain consistent levels of c-Jun during 10-week chronic denervation. Data normalized to WT 1 week post-injury. Two-way ANOVA with Sidak's multiple comparisons test; *p<0.05. n = 5. (**B**) Representative images showing c-Jun/Sox10 double immunofluorescence in transverse sections of WT and OE/+ uninjured and injured distal stumps. The graph shows quantitation by cell counting. The c-Jun labeling of Sox10-positive nuclei in the two genotypes is comparable at 1 week, but reduced at 10 weeks in WT nerves only. Two-way ANOVA with Tukey's multiple comparison test; ***p<0.001. n = 3. All numerical data represented as means ± SEM, all scale bars: 100 μm.

was similar to that in WT animals (*Figure 5D,E*). This does not measure regeneration, but indicates that a comparable number of DRG and motor neurons project through the normal uninjured peroneal nerve in the two mouse lines. Second, although c-Jun levels in WT and c-Jun OE/+ mice diverge after 10 weeks of denervation, they are high, and similar, early after injury, when the capacity of WT nerves to support regeneration is optimal. In line with this, the regeneration support provided by WT and c-Jun OE/+ nerves was similar early after injury. Thus, in back-filling experiments no significant difference was seen between WT and c-Jun OE/+ mice in the numbers of DRG or motor neurons that regenerated through acutely transected distal stumps (immediate repair) when the tracer was applied 2 weeks after transection/repair (*Figure 5F* and not shown). Similarly, comparable numbers of back-filled motor neurons were obtained in WT and c-Jun OE/+ mice when back-filling was used to quantify regeneration 5 days after sciatic nerve crush (*Figure 5G*).

Together, these experiments indicate that under the conditions prevailing in the distal stump of cut nerves, Schwann cells are unable to maintain high c-Jun levels in the long term. The resulting

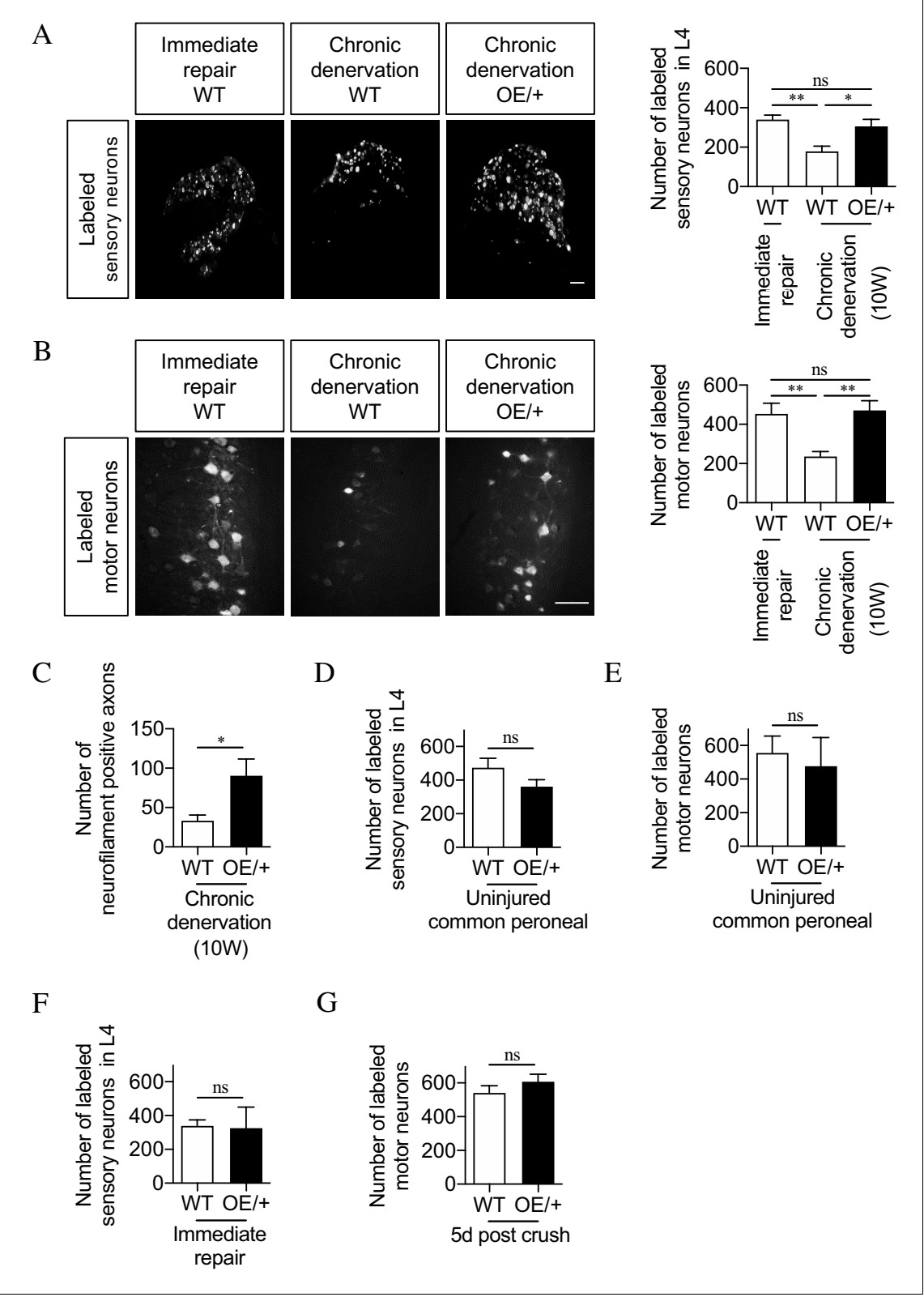

**Figure 5.** The regenerative capacity of c-Jun OE/+ nerves is maintained during chronic denervation. (**A**) Representative images showing Fluorogold-labeling of neurons in L4 DRGs of WT and c-Jun OE/+ mice after 2 weeks of regeneration into acutely transected (immediate repair) or chronically denervated (10 weeks) distal stumps. Quantification by cell counting is in the graph. The number of back-filled DRG neurons following regeneration through chronically denervated WT stumps was reduced, but maintained after regeneration through chronically denervated c-Jun OE/+ stumps. One-

*Figure 5 continued on next page*

*Figure 5 continued*

way ANOVA with Tukey's multiple comparison test; **p<0.005, *p<0.05, ns non-significant. WT immediate repair and chronic denervation n = 6, OE/+ chronic denervation n = 8. (B) Representative images showing Fluorogold- labeling of back-filled motor neurons in WT and c-Jun OE/+ mice after 2 weeks of regeneration into acutely transected (immediate repair) or chronically denervated distal stumps. Quantification is in the graph, showing that compared to immediate repair, the number of labeled neurons is reduced after regeneration through chronically denervated WT stumps, but not after regeneration through chronically denervated c-Jun OE/+ stumps. One-way ANOVA with Tukey's multiple comparison test; **p<0.005. WT immediate repair and chronic denervation n = 6, c-Jun OE/+ chronic denervation n = 8. (C) Counts of neurofilament+ immunofluorescent fibers in the distal stump 1 week after repair following chronic denervation of WT and c-Jun OE/+ nerves. In parallel experiments, the number of neurofilament+ fibers in WT nerves after immediate repair was about 200 (see *Figure 2D*). Unpaired Student's t-test; *p=0.0334. WT n = 4, c-JunOE/+ n = 3. (D, E) Counts of Fluorogold-labeled sensory (D) and motor (E) neurons in WT and c-Jun OE/+ mice following transection with immediate application of tracer. The number of back-filled sensory (p=0.1872) and motor (p=0.7153) neurons is similar. Unpaired Student's t-tests, n = 3 for each experimental condition. (F) The number of back-filled sensory neurons in WT and c-Jun OE/+ mice is similar after transection followed by immediate repair. Unpaired Student's t-test; p=0.9195. n = 3. (G) The number of labeled motor neurons in WT and c-Jun OE/+ mice is similar when tracer was applied 5 days after sciatic nerve crush. Unpaired Student's t-test; p=0.312. WT n = 5, c-Jun OE/+ n = 4. All numerical data represented as means ± SEM, all scale bars: 100 μm.

gradual reduction in c-Jun leads to deterioration of repair cell function, causing regeneration failure, a failure that can be corrected by restoring c-Jun levels.

## Improved regeneration in c-Jun OE+/-mice is unlikely to relate to altered cell numbers

To test whether the differential regeneration rates could be due to altered cell numbers, Schwann cell, macrophage and fibroblast nuclei were counted in tibial nerves by electron microscopy.

In experiments comparing young and aging animals, cells in the distal stump were counted 4 days after nerve cut without regeneration. Schwann cell and macrophage numbers and density were similar irrespective of genotype or age (*Figure 6A–C*). There was some decrease in fibroblast density in aged mice, but density and numbers were similar irrespective of genotype (*Figure 6D,E*). In aged mice, transverse profiles of the tibial nerve were larger mainly because of increased endoneurial connective tissue, but no difference was seen between the genotypes (*Figure 6F*).

In the experiments on chronic denervation, cells were counted in the distal stumps 2 and 10 weeks after cut without regeneration 5 mm from the injury site. There were ~250 Schwann cell nuclei in 2-week stumps, representing a ~2.5-fold increase from uninjured nerves (*Fazal et al., 2017*; *Figure 6G*). In the WT, Schwann cell numbers declined by ~30% during chronic denervation. This drop was prevented in c-Jun OE/+ nerves (*Figure 6G*). The density of macrophages was similar irrespective of genotype or length of denervation (*Figure 6H*), with numbers declining, irrespective of genotype, during chronic denervation (*Figure 6I*). Two weeks after injury, fibroblast density in WT and c-Jun OE/+ mice was similar. Fibroblast density was elevated in 10-week stumps of WT, but not c-Jun OE/+, mice (*Figure 6J*). Fibroblast numbers remained unchanged between genotypes following denervation (*Figure 6K*). Chronically denervated stumps of both genotypes had reduced nerve area, although this difference was less evident in c-Jun OE/+ nerves (*Figure 6L*).

These counts indicate that altered cell numbers are not a significant reason for improved regeneration in aged c-Jun OE/+ nerves. In chronically denervated nerves, the normal loss of Schwann cells in WT nerves is prevented in c-Jun OE/+ nerves. Even in 10-week WT stumps, however, cell numbers remain well above those in uninjured nerves (see Discussion).

## Shh signaling supports c-Jun expression

Seeking mechanisms that promote c-Jun expression in denervated Schwann cells, we considered signaling by Shh, a gene that is not expressed in developing Schwann cells or in intact nerves, but strongly upregulated in repair Schwann cells after injury (*Lu et al., 2000*; *Zhou et al., 2000*; *Arthur-Farraj et al., 2012*; *Hsin-Pin et al., 2015*). First, we analyzed Shh cKO mice, in which Shh is selectively inactivated in Schwann cells and in which nerve ultrastructure appears normal as expected (J Svaren unpublished). We found that in the mutants, c-Jun protein and phosphorylated c-Jun were decreased in the distal stump of 7-day transected sciatic nerve (*Figure 7A,B*). Levels of p75NTR

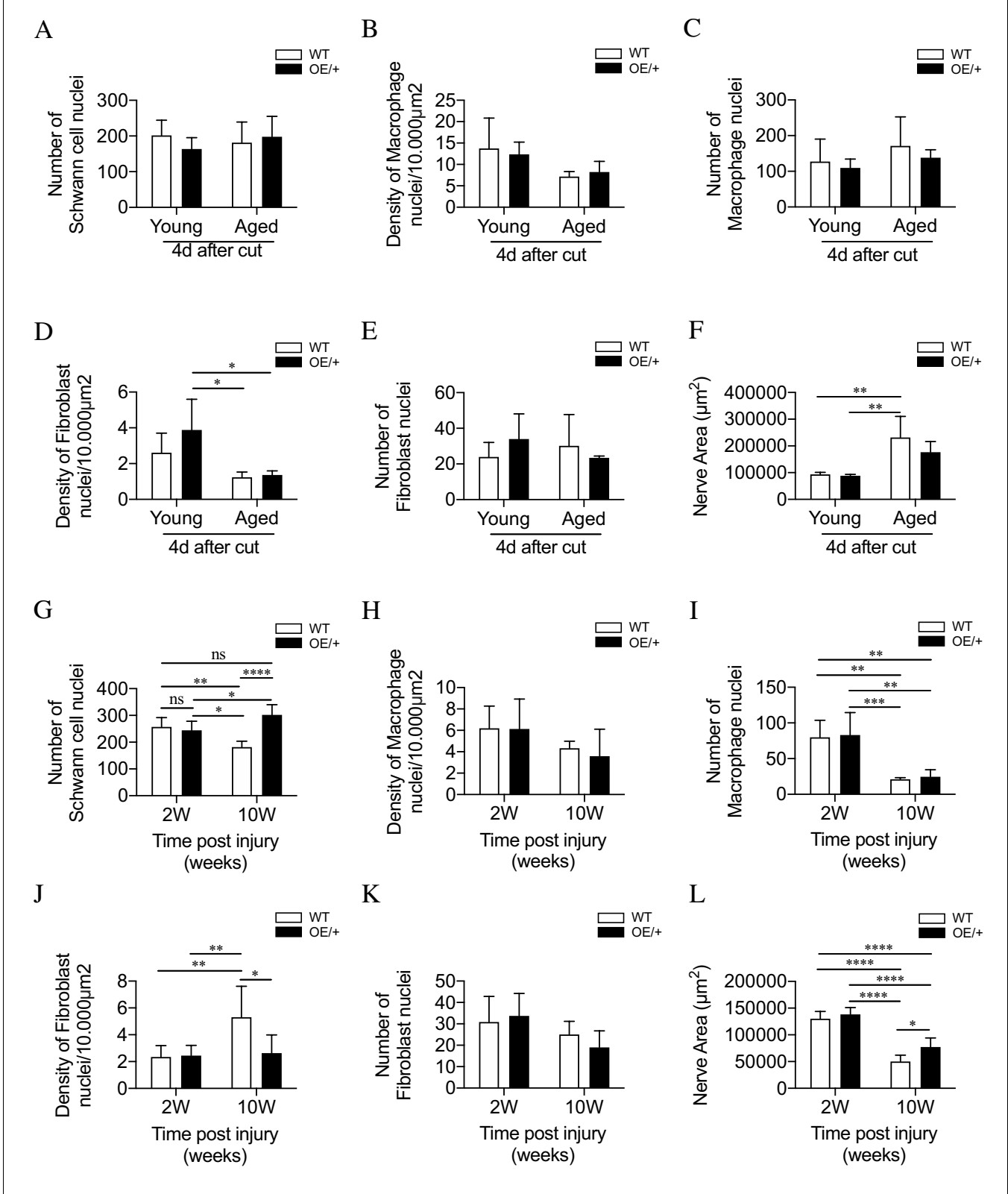

**Figure 6.** Cell number and nerve size in injured WT and c-Jun OE/+ nerves Cell nuclei were counted in whole transverse profiles of the tibial nerve, 5 mm from the injury site, using the electron microscope. (A) Schwann cell numbers in young and aged WT and c-Jun OE/+ nerves. (B) Macrophage density and (C) number in young and aged WT and c-Jun OE/+ nerves. (D) Fibroblast density and (E) number in young and aged WT and c-Jun OE/+ nerves: *p<0.05. (F) Whole transverse profiles of the tibial nerve were measured for the nerve area of young and aged WT and c-Jun OE/+ nerves;

*Figure 6 continued on next page*

*Figure 6 continued*

**p<0.005. For counts in A-F, n = 4 for each condition. (**G**) Schwann cell numbers in 2- and 10-week cut nerves of WT and c-Jun OE/+ mice; *p<0.05, **p<0.005, ****p<0.0001. (**H**) Macrophage density and (**I**) number in 2- and 10-week cut nerves of WT and c-Jun OE/+ mice; **p<0.005, ***p<0.001. (**J**) Fibroblast density and (**K**) number in 2- and 10-week cut nerves of WT and c-Jun OE/+ mice; *p<0.05, **p<0.005. (**L**) Whole transverse profiles of the tibial nerve were measured for the nerve area of 2- and 10-week cut nerves of WT and c-Jun OE/+ mice; *p<0.05, ****p<0.0001. For counts in G-L, 2-week WT and c-Jun OE/+ n = 7, 10-week WT and c-Jun OE/+ n = 5. All number data represented as means ± SEM analyzed by two-way ANOVA with Tukey's multiple comparison test.

The online version of this article includes the following source data for figure 6:

**Source data 1.** Cell counts and measurements relating to *Figure 6*.

protein, which is positively regulated by c-Jun in Schwann cells (*Arthur-Farraj et al., 2012*), were also reduced 7 days after injury in the mutants (*Figure 7C*). Substantiating these observations, two Shh signaling agonists, SAG and purmorphamine, upregulated c-Jun protein in purified Schwann cell cultures (*Figure 7D,E*). This was confirmed using c-Jun/Sox10 double immunolabeling (*Figure 7F*). SAG also increased the expression of two c-Jun target genes that promote nerve regeneration *Bdnf* and *Gdnf* (*Figure 8A,B*). Further, SAG promoted another effect of c-Jun, that of enhancing the elongated bi- or tri-polar shape in vitro that reflects the elongation and branching of repair cells in vivo (*Arthur-Farraj et al., 2012*; *Gomez-Sanchez et al., 2017*; *Figure 8C*).

After injury, Shh-dependent enhancement of c-Jun is likely to be mediated by Shh derived from Schwann cells, which are the major source of Shh in injured nerves. In support of such an autocrine hedgehog signaling loop, cyclopamine alone, which blocks cellular responses to Shh, downregulated c-Jun protein and sharply suppressed c-Jun phosphorylation in cultured Schwann cells (*Figure 8D*).

Together these observations show that Shh promotes Schwann cell c-Jun expression in vitro and in vivo, and support the idea that injury triggers an autocrine Shh signaling loop to elevate c-Jun in repair cells.

## Analysis of gene expression in distal nerve stumps of WT and c-Jun OE/+ mice

Young and aging mice.

RNA sequence analysis was performed on uninjured and 3-day cut sciatic nerves of young (6–8 weeks) and aged (11–12 months) WT mice, and aged c-Jun OE/+ mice. Global gene expression was analyzed comparing (i) uninjured nerves, (ii) 3-day cut nerves, and (iii) the injury response (3-day cut vs uninjured) in young and aged mice (*Figure 9—figure supplement 1A*).

In uninjured WT nerves, out of 15,995 genes present, 1477 genes were differentially expressed between young and aged mice. Of these, 1154 were upregulated and 323 downregulated in aging mice compared to young ones (FC >2; and FDR 0.05) (*Supplementary file 1 A*; *Supplementary file 2 A*). We tested whether the 173 genes we previously identified as c-Jun-regulated injury genes (*Arthur-Farraj et al., 2012*) were implicated in these age-dependent differences. In the present data set, 138 of the 173 genes were present (*Supplementary file 3*). They showed a strong enrichment among the 1477 genes differentially expressed between young and aging mice (*Figure 9A*).

In 3-day cut WT nerves, of 17,334 genes present, 398 genes were differentially expressed between young and aging mice. Of these, 268 were upregulated and 130 downregulated in aging nerves (*Supplementary file 1 B*; *Supplementary file 2 B*). This gene set contains candidate genes responsible for the difference in regeneration support provided by young and aged Schwann cells (*Painter et al., 2014*) and present experiments. In agreement with *Painter et al., 2014*, trophic factors such as GDNF, BDNF, NGF, erythropoietin, and FGF were not among the differentially expressed genes. This suggests that expression of trophic factors often implicated in regeneration may not explain different regeneration between young and aged mice. The 138 c-Jun-regulated injury genes (*Supplementary file 3*; *Arthur-Farraj et al., 2012*) were highly enriched among the age-regulated genes (*Figure 9A*).

Examining the injury response (3-day cut vs uninjured nerve), 822 genes showed significant difference when young and aging WT mice were compared (*Supplementary file 1 C*; *Supplementary file 2 C*). The 138 c-Jun-regulated injury genes were strongly enriched among these 822 genes (*Figure 9A*).

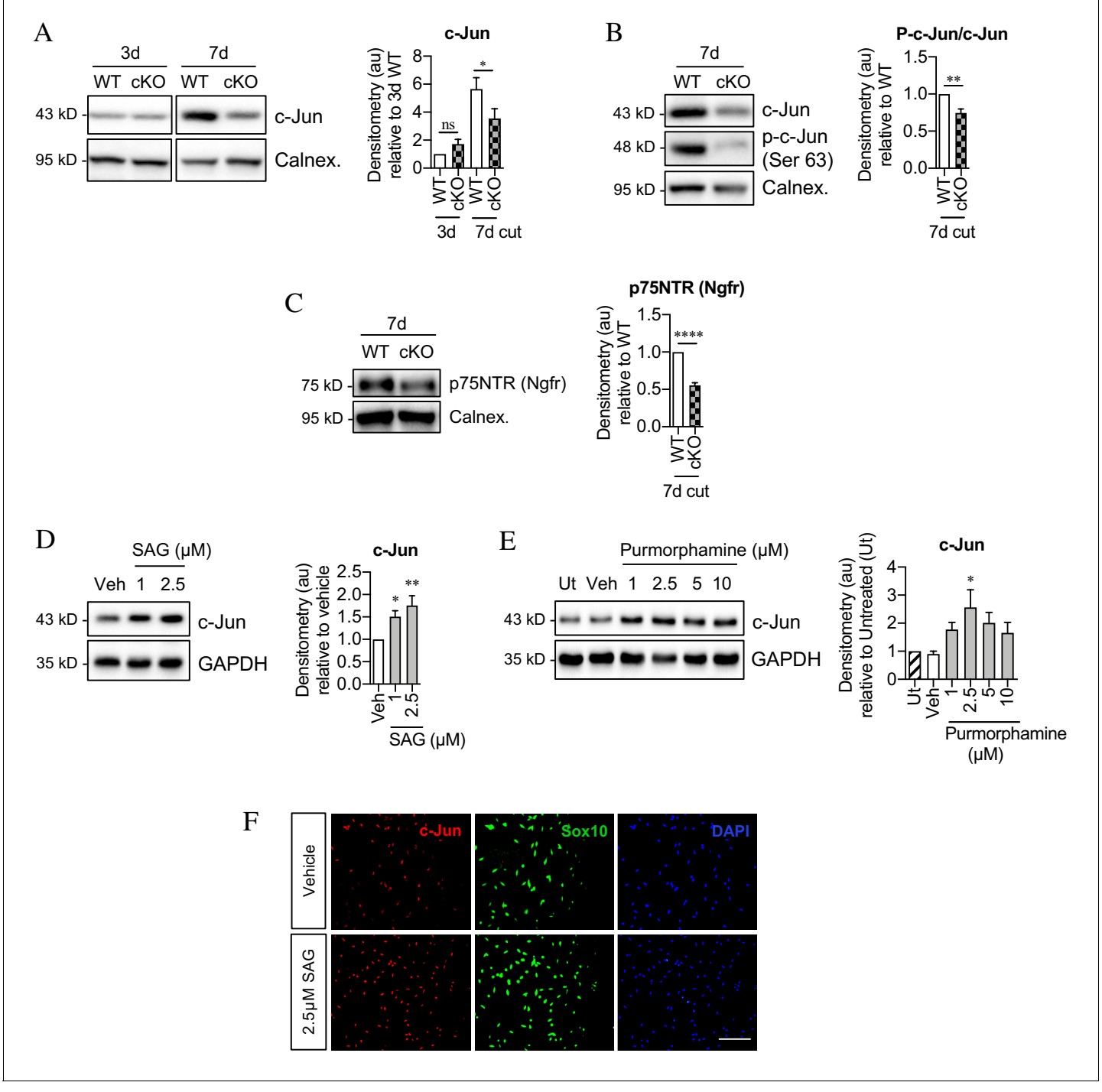

**Figure 7.** Sonic hedgehog promotes c-Jun activation in Schwann cells in vivo and in vitro. (**A**) Representative western blot showing c-Jun expression in WT and Shh cKO (cKO) nerves 3 and 7 days after cut. Quantitation is shown in the graph where the data are normalized to WT 3 days post-cut. Two-way ANOVA with Sidak's test; *p<0.05. n = 5 for each genotype. (**B**) Representative Western blot showing c-Jun and phosphorylated c-Jun (p-c-Jun) in WT and Shh cKO distal nerve stumps 7 days post-cut. Quantitation is shown in the graph where the data are normalized to WT 7 days post-cut. Unpaired Student's t-test; **p=0.0014, n = 5 for each genotype. (**C**) Representative western blot showing p75NTR protein in WT and Shh cKO nerves 7 days post-cut. The graph shows quantitation of the results. Data are normalized to WT 7 days post-injury. Unpaired Student's t-test; ****p=<0.0001. n = 5 for each genotype. (**D**) Representative western blot showing c-Jun in Schwann cell cultures exposed to SAG for 48 hr. (Veh: DMSO vehicle). Quantitation is shown in the graph where the data are normalized to vehicle. One-way ANOVA with Dunnet's test; *p<0.05, **p<0.005. n = 6. (**E**) Representative western blot showing c-Jun in Schwann cell cultures exposed to purmorphamine for 48 hr. Quantitation is shown in the graph where the

*Figure 7 continued on next page*

*Figure 7 continued*

data are normalized to vehicle. One-way ANOVA with Dunnet's test; *p<0.05. n = 3. (**F**) Representative immunofluorescence images showing increased c-Jun labeling of Sox10-positive Schwann cell nuclei after 48 hr incubation with SAG compared to vehicle (DMSO). Scale bar: 100 μm.

Gene set enrichment analysis (GSEA) on the above conditions showed that the 138 c-Jun genes were highly enriched among the genes downregulated in aging uninjured nerves and in aging 3-day cut nerves (*Figure 9—figure supplement 1B,C*). When the injury response (3-day cut vs uninjured) of young and aging WT mice was analyzed, the strongest enrichment of c-Jun genes was observed in genes upregulated in young nerves (*Figure 9—figure supplement 1D*). While c-Jun genes are also upregulated in aged nerves post-injury, their enrichment was not as high as in young nerves.

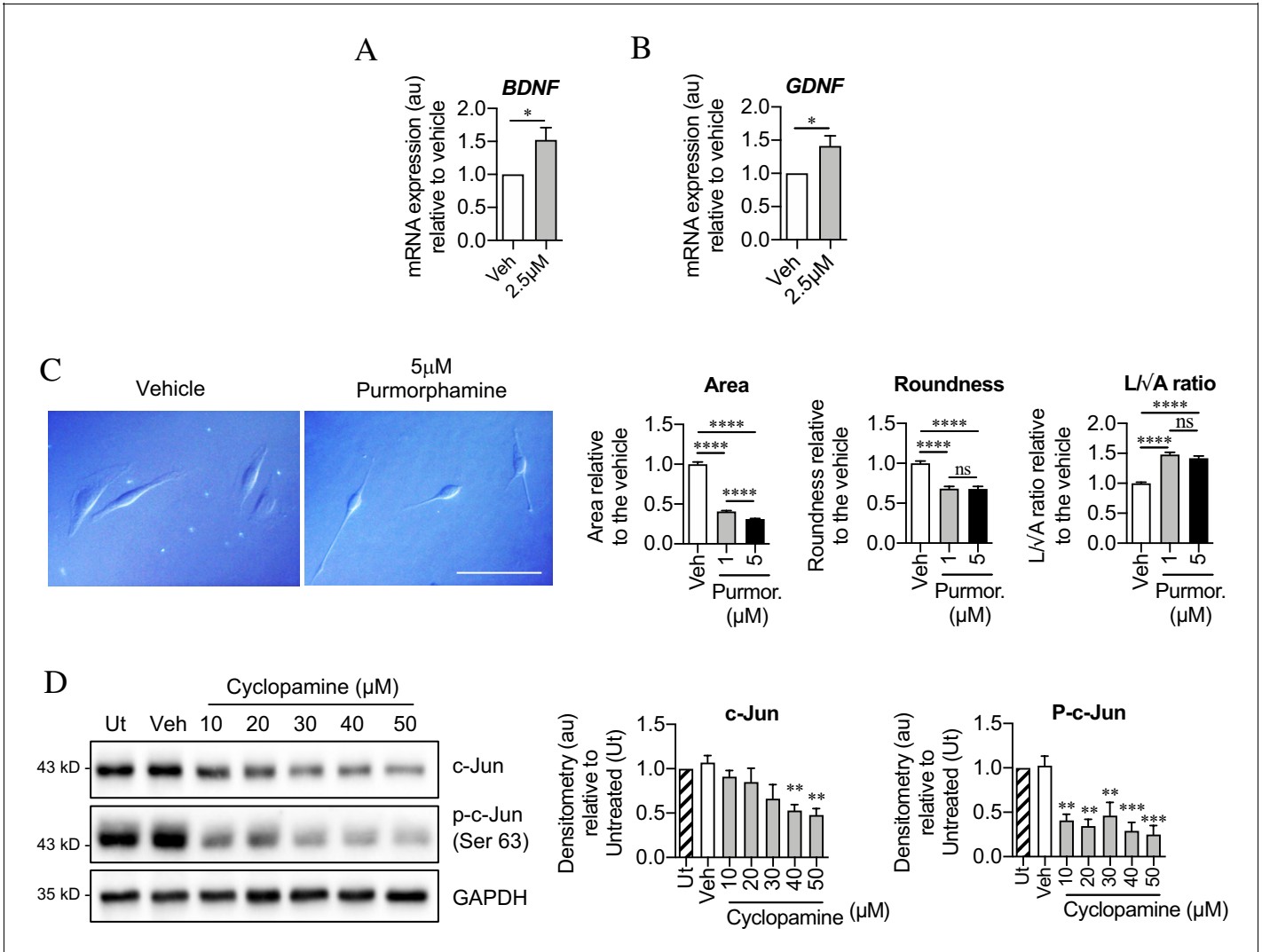

**Figure 8.** Sonic hedgehog plays a role in c-Jun activation in Schwann cells via autocrine signaling. (**A, B**) qPCR showing mRNA expression of (**A**) *Bdnf* *p=0.0314 and (**B**) *Gdnf* *p=0.0382 in Schwann cell cultures incubated for 48 hr with SAG. Data normalized to vehicle. Unpaired Student's t-tests. n = 4 for each condition. (**C**) Differential interference contrast (DIC) microscopy showing changes in Schwann cell morphology after 48 hr incubation with purmorphamine (DMSO vehicle). Scale bar: 50 μm. Graphs depict changes in cell area, roundness and length/√area following incubation with purmorphamine, demonstrating enhancement of elongated morphology. One-way ANOVAs with Tukey's multiple comparison test ***p<0.001. n = 3, each experiment involving measurement of 100 cells per condition. (**D**) Representative Western blots showing c-Jun and phosphorylated c-Jun in cultured Schwann cells after 48 hr incubation with cyclopamine alone (DMSO vehicle). The graphs show quantitation of the blots. Data are normalized to vehicle. One-way ANOVA with Dunnet's multiple comparisons test; **p<0.005; ***p<0.001. n = 3. All numerical data represented as means ± SEM.

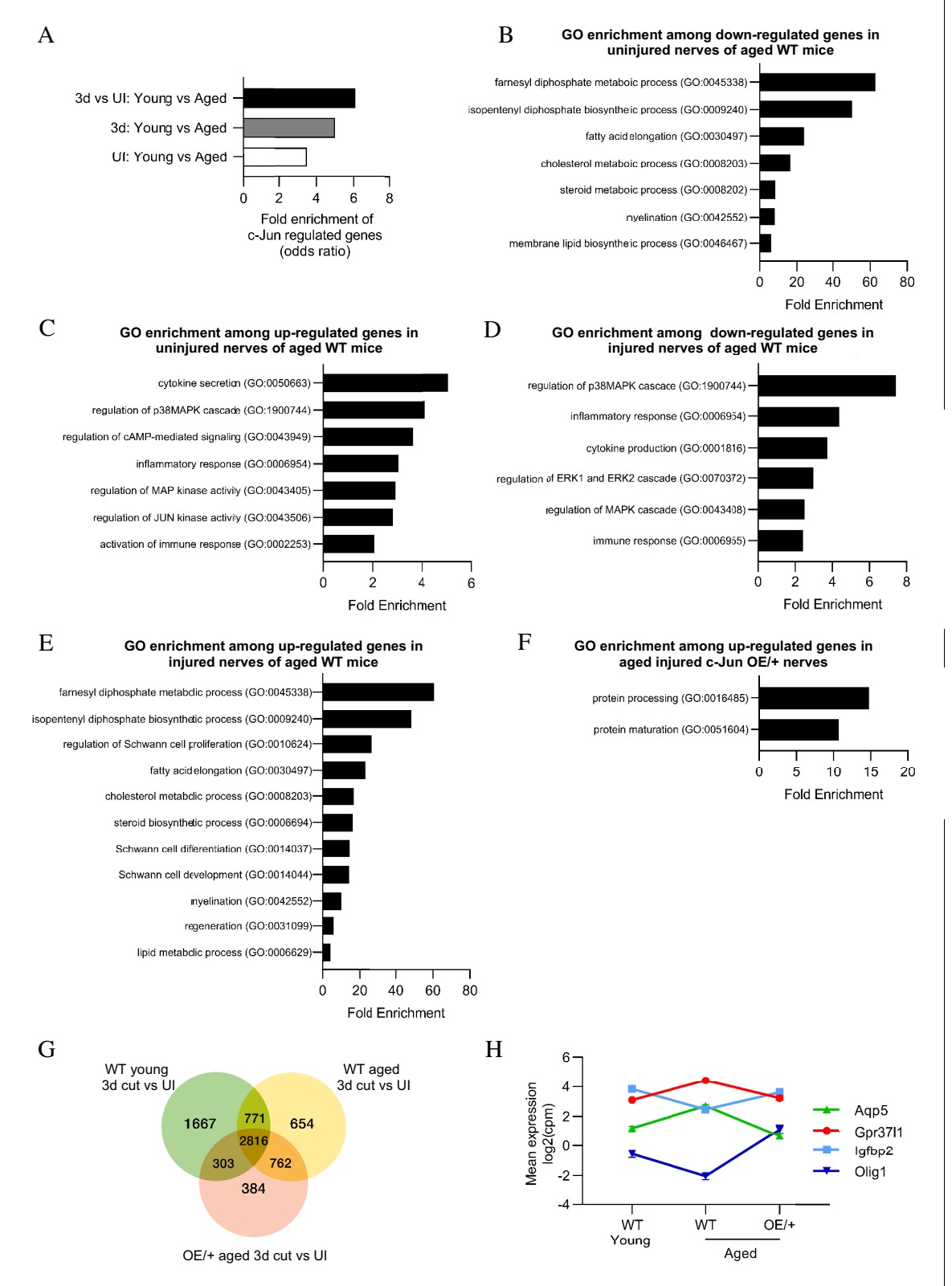

**Figure 9.** Bioinformatics analysis of RNA seq. data from young and aged nerves. (**A**) Over-representation analysis showing enrichment of c-Jun-regulated genes in various WT injury paradigms. p=3.2×10$^{-8}$ for UI young vs aged; p=1×10 x $^{-7}$ for 3-day cut young vs aged; p=2.3×10$^{-13}$ for the injury response. p-Values computed by one-sided Fisher's exact test. (**B**) GO terms downregulated and (**C**) upregulated in uninjured nerves of aged WT mice (absolute fold change >2 and FDR < 0.05). (**D**) GO terms downregulated and (**E**) upregulated in the injury response of aged WT mice (absolute

*Figure 9 continued on next page*

*Figure 9 continued*

fold change >2 and FDR < 0.05). (F) When aged c-Jun OE/+ and WT nerves are compared, genes associated with protein processing (FDR = 0.00318) and maturation (FDR = 0.0153) are significantly enriched in aged c-Jun OE/+ nerves compared to aged WT. (G) Venn diagram showing numbers of differentially expressed genes between young and aged 3-day cut WT nerves and aged 3-day cut OE/+ nerves, compared to their uninjured counterparts. (H) Mean expression of four c-Jun-regulated genes with significantly different expression between young and aged WT nerves but not between young WT and aged c-Jun OE/+ nerves(absolute fold change >2 and FDR < 0.05).

The online version of this article includes the following figure supplement(s) for figure 9:

**Figure supplement 1.** Bioinformatics analysis of aged and young nerves following injury.

These correlations between enrichment of c-Jun-regulated genes and Schwann cell age suggest that the c-Jun-regulated repair program is disproportionately vulnerable during the aging process.

Gene ontology (GO) analysis showed that in aged uninjured WT nerves, downregulated genes were largely involved in lipid metabolism, as well as myelination, while genes involved in the immune system were prominent among those upregulated (*Figure 9B,C*; reviewed in *Melcangi et al., 1998*; *Melcangi et al., 2000*; *Büttner et al., 2018*). Similar analysis of the injury response (3-day cut vs uninjured) showed reduced activation of immune genes in aged WT nerves (*Scheib and Höke, 2016*; *Büttner et al., 2018*). In aged nerves, MAPK pathways were also suppressed while lipid metabolism and Schwann cell differentiation genes were enhanced (*Figure 9D,E*). Together this indicates suppressed Schwann cell reprogramming and repair cell activation in nerves of older WT mice. Testing the effects of enhanced c-Jun expression on the aged injury response, we found that pathways associated with protein processing and maturation were upregulated in aged c-Jun OE/+ nerves compared with aged WT nerves (*Figure 9F*).

To further determine genes that may contribute to the restoration of regeneration in aged c-Jun OE/+ mice, the injury responses in young and aged WT mice and aged c-Jun OE/+ mice were compared (*Figure 9G*). Of particular interest are the 303 genes that show significant injury response in young WT mice but not in aging WT mice, but are again significantly regulated by restoring c-Jun to youthful levels in aging c-Jun OE/+ mice (*Supplementary file 2 D*).

Among the 138 c-Jun-regulated genes, we looked for a correlation between a failure and restoration of gene expression on the one hand, and failure and restoration of regeneration on the other. In 3-day cut aged WT nerves, where regeneration fails, 16 c-Jun-regulated genes were differentially expressed compared to 3-day cut young WT nerves. Four of these, *Aqp5*, *Gpr37L1*, *Igfbp2*, and *Olig1*, were restored in aged c-Jun OE/+ nerves, where regeneration is restored (*Figure 9H*). Thus, in aging mice, both regeneration failure and the expression defect of these four genes was restored to levels in young mice, by elevating c-Jun levels.

Chronic denervation.

Gene expression was examined in uninjured nerves and in 1 and 10-week cut sciatic nerves of WT and c-Jun OE/+ mice (*Figure 10—figure supplement 1A*). Expression of 1581 genes changed significantly during chronic denervation (*Supplementary file 4*; *Supplementary file 2 E*). In 10-week cut nerves, 601 of these genes were downregulated, including genes associated with repair cells such as *Gdnf*, *Shh*, and *Ngfr*, while 980 genes were upregulated. The 138 c-Jun-regulated genes showed a highly significant 5.8-fold enrichment (p=2.2×10$^{-16}$) among the 1581 genes regulated during chronic denervation. GSEA enrichment analysis showed that c-Jun genes were some of the most downregulated genes during chronic denervation. (*Figure 10—figure supplement 1B*).

GO analysis showed that the major genes downregulated during chronic denervation in WT nerves involved the cell cycle, DNA replication, and repair. Glial cell differentiation genes and MAPK pathways, potential activators of c-Jun, were also suppressed (*Figure 10A*). Chronic denervation involved a prominent upregulation of neuro-glia signaling genes (chiefly related to GABA and adrenergic signaling), but also regulators of differentiation, Notch and cAMP signaling (*Figure 10B*). To test the effects of maintaining c-Jun protein levels during the 10-week chronic denervation, we identified genes differentially expressed between 10-week cut WT and c-Jun OE/+ nerves (*Figure 10C*). This showed strong upregulation, in c-Jun OE/+ mice, of pathways involved in PNS and Schwann cell development and differentiation.

To further determine genes that may contribute to the restoration of regeneration in aged c-Jun OE/+ mice, the injury response in the three situations analyzed in the regeneration experiments, WT

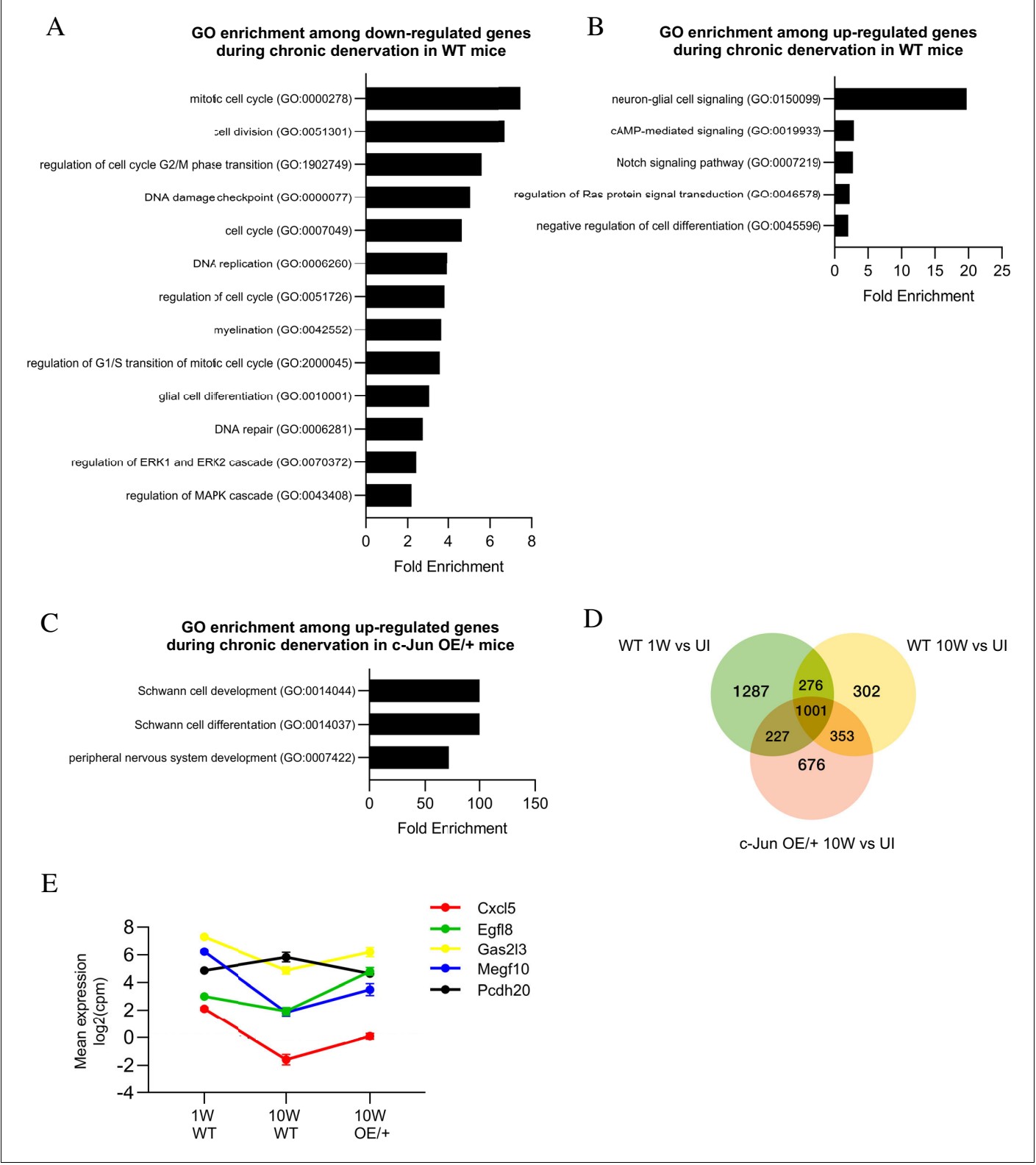

**Figure 10.** Bioinformatics analysis of RNA seq. data from acutely and chronically denervated nerves. (**A**) GO terms downregulated and (**B**) upregulated in WT nerves during chronic denervation. (**C**) When chronically denervated c-Jun OE/+ and WT nerves were compared, GO terms associated with Schwann cell differentiation (FDR = 0.00397) and PNS development (FDR = 0.0173) were enriched in c-Jun OE/+ nerves. (**D**) Venn diagram showing numbers of differentially expressed genes between WT nerves following acute (1 week) and chronic (10-week) denervation and c-Jun OE/+ nerves
*Figure 10 continued on next page*

*Figure 10 continued*

following chronic denervation, compared to their uninjured counterparts. (E) Mean expression of five c-Jun-regulated genes with significantly different expression between acute and chronic WT nerves, but not between acute WT and chronic c-Jun OE/+ nerves (absolute fold change >2 and FDR < 0.05).

The online version of this article includes the following figure supplement(s) for figure 10:

**Figure supplement 1.** Bioinformatics analysis of nerves after chronic injury.

1 week, WT 10 weeks, and c-Jun OE/+ 10 weeks, was compared (*Figure 10D*). A point of interest are the 227 genes that showed an injury response in WT 1-week nerves and in c-Jun OE/+ 10-week nerves, both of which show fast regeneration, but no injury response in WT 10-week nerves, where regeneration is slow (*Supplementary file 2 F*).

As when studying aged mice, we looked among the 138 c-Jun-regulated injury genes for candidates involved in decreased regeneration in 10-week cut WT nerves and the restoration of regeneration in 10-week cut c-Jun OE/+ nerves. Fifty of the 138 genes changed expression during chronic denervation in WT nerves, where regeneration is poor. In chronically denervated c-Jun OE/+ nerves, where regeneration is restored, expression levels were restored, completely or partially, in the case of five of these genes, *Cxcl5*, *Egfl8*, *Gas213*, *Megf10*, and *Pcdh20* (*Figure 10E*). These correlations provide a basis for considering these genes as candidates down-stream of c-Jun for involvement in the restoration of regeneration in chronically denervated nerves of c-Jun OE/+.

## Discussion

The present results indicate that reduced expression of c-Jun is an important factor in the repair cell failures seen during aging and chronic denervation. In both situations, Schwann cells of injured nerves fail to achieve or maintain high c-Jun levels, and in both cases, correction of c-Jun expression restores regeneration deficits. This highlights the importance of c-Jun in the function of repair Schwann cells, provides a common molecular link between two apparently unrelated problems in nerve repair, and points to manipulation of c-Jun-regulated pathways as a potential route for improving the outcome of nerve injuries.

By using neuron back-filling, this study provides a direct quantitative measure of neuronal regeneration capacity in vivo and how this is controlled by Schwann cell c-Jun levels. It also opens new questions that remain to be investigated. In particular, to what extent does c-Jun in Schwann cells determine other factors that are also important for repair. This includes the length of time neurons are able to sustain axon growth after injury, sprouting, axonal misrouting, targeting, and synapse reformation.

A previously identified gene set regulated by c-Jun in injured nerves (*Arthur-Farraj et al., 2012*) was found to be highly enriched among the genes affected by aging or chronic denervation in WT mice. The expression of a small group of genes was also positively correlated both with c-Jun levels and regeneration, suggesting that their role in Schwann cells or other cells in the nerve merits further study. In aging mice, this encompasses *Aqp5*, *Gpr37L1*, *Igfbp2*, and *Olig1*, all of which have been studied in glial cells. Igfbp2 promotes phosphorylation of Akt, a pathway that is linked to Schwann cell proliferation and differentiation (reviewed in *Ma et al., 2015*; *Boerboom et al., 2017*; *Jessen and Arthur-Farraj, 2019*). Gpr37L1 is a receptor for prosaposin and prosapeptide (*Meyer et al., 2013*). In Schwann cells, prosapeptide phosphorylates MAPK (*Hiraiwa et al., 1997*) and prosaposin is secreted after nerve injury, facilitating regeneration (*Hiraiwa et al., 1999*). In experiments on chronic denervation, this gene group encompasses *Cxcl5, Egfl8, Gas2I3, Megf10*, and *Pcdh20*. All these genes were previously shown to be upregulated in Schwann cells after injury (*Zhang et al., 2011*; *Tanaka et al., 2013*; *Weiss et al., 2016*; reviewed in *Ma et al., 2016*; *Brosius Lutz et al., 2017*). Cxcl5 activates STAT3 (*Zhang et al., 2011*), a transcription factor that we have shown to be important for maintaining repair cells during chronic denervation (*Benito et al., 2017*). Gas2I3 has a role in the cell cycle, and Megf10 in phagocytosis (*Wolter et al., 2012*; *Chung et al., 2013*).

Since c-Jun levels in injured nerves are a major determinant of effective repair, it is important to identify signals that control c-Jun expression. The present results suggest that Shh has a role in this

process. In injured nerves of Shh cKO mice, there is reduced c-Jun activation and diminished Schwann cell expression of the c-Jun target p75NTR. In purified Schwann cells, application of Shh elevates c-Jun, while cyclopamine alone suppresses c-Jun. Shh also promotes Schwann cell elongation. Further, during chronic denervation, Shh expression, like that of c-Jun, is substantially reduced. Previous work also implicates Shh signaling in repair. Shh is upregulated in Schwann cells after injury (*Hashimoto et al., 2008*; *Arthur-Farraj et al., 2012*; *Yamada et al., 2018*), and exposure to Shh improves nerve regeneration in various settings (*Pepinsky et al., 2002*; *Bond et al., 2013*; *Martinez et al., 2015*; *Yamada et al., 2018*). Inhibition of Shh signaling reduces Schwann cell expression of BDNF, motor neuron survival after injury and axon regeneration (*Hashimoto et al., 2008*; *Yamada et al., 2020*), and a molecular link between Shh signaling and Jun activation has been established in various cell lines (*Laner-Plamberger et al., 2009*; *Kudo et al., 2012*). Further in vivo experiments using Shh cKO mice as well as Shh agonists and antagonists are needed to conclusively determine the involvement of Shh in regeneration. At present, however, the data presented here and previous work are consistent with the existence of an autocrine Shh signaling loop activated by injury to promote expression of c-Jun and the repair cell phenotype.

In c-Jun OE/+ mice, we considered whether restoration of c-Jun levels altered cell numbers, thus promoting regeneration. In aging mice, the results appear to exclude this, since cell numbers in the mutant and the WT are similar. During chronic denervation, Schwann cell numbers remain constant in c-Jun OE/+mice, but fall by about 30% in the WT. Since there is now evidence that Schwann cell proliferation may not be essential for regeneration, contrary to common assumptions, the relationship between cell numbers and repair is currently unclear (*Kim et al., 2000*; *Atanasoski et al., 2001*; *Yang et al., 2008*; for discussion see *Jessen and Mirsky, 2019*). Even in WT mice, cell number after chronic denervation remains nearly twice that in uninjured nerves. It is therefore unlikely that the changes in Schwann cell numbers are the key reason for the reduced regeneration support provided by 10-week cut WT stumps, or the increase in support provided by 10-week cut c-Jun OE/+ stumps.

The degree of reduction in transverse nerve area after chronic denervation could also affect repair. However, the area of 10-week cut c-Jun OE/+ nerves, while increased compared to 10-week cut WT, remains ~50% smaller than that of 2-week cut WT nerves. Nevertheless, regeneration through these nerves is similar. The relationship between nerve area and regeneration in these experiments may therefore not be straightforward.

During longer denervation times, cell loss and nerve shrinking will increasingly impede repair. These slow, atrophic changes, likely involving regulation of cell death and proliferation, have not been extensively studied, although STAT3 has recently been implicated in the long-term maintenance of repair cells (*Benito et al., 2017*). Previously, c-Jun was shown to influence both apoptosis and proliferation in repair Schwann cells (*Parkinson et al., 2001*; *Parkinson et al., 2004*; *Parkinson et al., 2008*), but the particular way in which c-Jun levels determine nerve atrophy remains to be determined.

It has become clear that the injury-induced reprogramming of Schwann cells to cells specialized to support nerve regeneration is regulated by dedicated mechanisms, including c-Jun, STAT3, merlin, and H3K27 trimethylation-related epigenetic controls, that operate selectively in repair cells, and have a relatively minor or undetectable function in Schwann cell development (reviewed in *Jessen and Mirsky, 2019*). The present work provides evidence that an impairment of one of these mechanisms, c-Jun, is a major contributor to two major categories of regeneration failure, aging and chronic denervation. It will be important to extend this study to other regulators of repair cells as a basis for developing molecular interventions for promoting repair in the PNS.

## Materials and methods

**Key resources table**

| Reagent type (species) or resource | Designation | Source or reference | Identifiers | Additional information |
|---|---|---|---|---|
| *Mpz < Cre/+>;* *Rosa26c-Junstopf < f/+>,* C57BL/6J background, *Mus musculus* both sexes used | c-Jun OE/+ mouse | *Fazal et al., 2017* | RRID:MGI: | |

*Continued on next page*

*Continued*

| Reagent type (species) or resource | Designation | Source or reference | Identifiers | Additional information |
|---|---|---|---|---|
| *Mpz < Cre/+>; Jun < f/+>,* C57BL/6J background, *Mus musculus* both sexes used | c-Jun cKO mouse | *Arthur-Farraj et al., 2012* | *Jun^{tm4Wag}* | RRID:MGI:2445420 |
| *Mpz < Cre/+>; Shh < f/+>,* C57BL/6J background, *Mus musculus* both sexes used | Shh cKO mouse | Jackson Laboratory | B6;129-*Shh^{tm2Amc}*/J | RRID:IMSR_JAX:004293 |
| *Mpz < Cre/+>,* C57BL/6J background, *Mus musculus* both sexes used | Mpz-Cre mouse | Jackson Laboratory | B6N.FVB-Tg (*Mpz-cre)26Mes*/J; | RRID:IMSR_JAX:017927 |
| Antibody | Anti- c-Jun (rabbit monoclonal) | Cell Signaling | Cat #9165; RRID:AB_2130165 | WB (1:1000) IF (1:800) |
| Antibody | Anti- P-c-Jun (rabbit polyclonal) | Cell Signaling | Cat#9261; RRID:AB_2130162 | WB (1:1000) |
| Antibody | Anti- p75NTR (Ngfr) (rabbit polyclonal) | Millipore | Cat#AB1554; RRID:AB_90760 | WB (1:1000) |
| Antibody | Anti- GAPDH (rabbit polyclonal) | Sigma-Aldrich | Cat#G9545; RRID:AB_796208 | WB (1:5000) |
| Antibody | Anti- Canelxin (rabbit polyclonal) | Enzo Life Sciences | Cat#ADI-SPA-860-D; RRID:AB_2038898 | WB (1:1000) |
| Antibody | Anti- sox10 (goat polyclonal) | R and D Systems | Cat#AF2864; RRID:AB_442208 | IF (1:100) |
| Antibody | Anti- CGRP (rabbit monoclonal) | Peninsula Laboratories | Cat#T-4032; RRID:AB_518147 | IF (1:1000) |
| Antibody | Anti- Neurofilament (chicken polyclonal) | Abcam | Cat#ab4680; RRID:AB_304560 | IF (1:1000) |
| Antibody | Anti- Rabbit IgG, HRP-linked (Goat polyclonal) | Cell Signaling | Cat#7074; RRID:AB_2099233 | WB (1:2000) |
| Antibody | Cy3 anti-Rabbit IgG (H+L) (Donkey polyclonal) | Jackson Immuno Research Labs | Cat#711-165-152; RRID:AB_2307443 | IF (1:500) |
| Antibody | Anti-Goat Alexa 488 Conjugated (Donkey polyclonal) | Molecular Probes - Thermo Fisher | Cat#A11057; RRID:AB_2534104 | IF (1:1000) |
| Antibody | Anti-Rabbit Alexa 488 Conjugated (Donkey polyclonal) | Molecular Probes - Thermo Fisher | Cat#A11008; RRID:AB_143165 | IF (1:1000) |
| Antibody | Anti-Chicken Alexa 488 Conjugated (Goat polyclonal) | Molecular Probes - Thermo Fisher | Cat#A-11039; RRID:AB_2534096 | IF (1:1000) |
| Sequence-based reagent | *Bdnf*_F | *Benito et al., 2017* | PCR primers | TCATACTTCGGTTGCATGAAGG |
| Sequence-based reagent | *Bdnf*_R | *Benito et al., 2017* | PCR primers | AGACCTCTCGAACCTGCCC |
| Sequence-based reagent | *c-Jun*_F (Cells) | *Benito et al., 2017* | PCR primers | AATGGGCACATCACCACTACAC |
| Sequence-based reagent | *c-Jun*_R (Cells) | *Benito et al., 2017* | PCR primers | TGCTCGTCGGTCACGTTCT |
| Sequence-based reagent | *c-Jun*_F (Tissue) | *Benito et al., 2017* | PCR primers | CCTTCTACGACGATGCCCTC |
| Sequence-based reagent | *c-Jun*_R (Tissue) | *Benito et al., 2017* | PCR primers | GATTCGGGCCACTTGGAGTT |
| Sequence-based reagent | *Gdnf*_F | *Benito et al., 2017* | PCR primers | GATTCGGGCCACTTGGAGTT |
| Sequence-based reagent | *Gdnf*_R | *Benito et al., 2017* | PCR primers | GACAGCCACGACATCCCATA |
| Sequence-based reagent | *Calnexin*_F | *Benito et al., 2017* | PCR primers | CAACAGGGGAGGTTTATTTTGCT |
| Sequence-based reagent | *Calnexin*_R | *Benito et al., 2017* | PCR primers | TCCCACTTTCCATCATATTTGGC |

*Continued on next page*

*Continued*

| Reagent type (species) or resource | Designation | Source or reference | Identifiers | Additional information |
|---|---|---|---|---|
| Sequence-based reagent | *Gapdh*_F | *Benito et al., 2017* | PCR primers | AGGTCGGTGTGAACGGATTTG |
| Sequence-based reagent | *Gapdh*_R | *Benito et al., 2017* | PCR primers | TGTAGACCATGTAGTTGAGGTCA |
| Sequence-based reagent | *Mpz*_F | *Benito et al., 2017* | PCR primers | GCTGGCCCAAATGTTGCTGG |
| Sequence-based reagent | *Mpz*_R | *Benito et al., 2017* | PCR primers | CCACCACCTCTCCATTGCAC |
| Commercial assay or kit | Kapa mRNA HyperPrep Kit | Roche | Cat#KK8581, 08098123702 | |
| Commercial assay or kit | RNeasy Micro Extraction Kit | Qiagen | Cat#74004 | |
| Chemical compound, drug | Purmorphamine | Sigma-Aldrich | Cat#540220 | Concentration: various, see figures |
| Chemical compound, drug | Smoothened Agonist (SAG) | Merck-Sigma-Aldrich-Calbiochem | Cat#566660 | Concentration: various, see figures |
| Chemical compound, drug | Cyclopamine | Merck-Sigma-Aldrich-Calbiochem | Cat#CAS 4449-51-8 | Concentration: various, see figures |
| Software, algorithm | Samtools version 1.2 | *Li et al., 2009* | RRID:SCR_002105 | |
| Software, algorithm | Picard tools version 1.140 | http://broadinstitute.github.io/picard/ | RRID:SCR_006525 | |
| Software, algorithm | featureCounts | *Liao et al., 2014* | RRID:SCR_012919 | |
| Software, algorithm | edgeR | *Robinson et al., 2010* | RRID:SCR_012802 | |
| Software, algorithm | Gen ser enrichment analysis (GSEA) | *Subramanian et al., 2005* | RRID:SCR_003199 | |
| Software, algorithm | Gen ontology (GO) analysis – PANTHER classification system | *Mi et al., 2013* | RRID:SCR_004869 | |
| Software, algorithm | GraphPad Prism 9.0.0 | GraphPad Prism | RRID:SCR_002798 | |
| Software, algorithm | Bio Rad ChemiDoc MP Imaging System | Bio Rad | RRID:SCR_019037 | |
| Other | Fluorogold | Fluorochrome | Fluoro-gold 20 mg | Made up to 4% |
| Other | DAPI | Thermo Fisher | Cat#D1306 | IF (1:40,000) |

## Transgenic mice

Animal experiments conformed to UK Home Office guidelines under the supervision of University College London (UCL) Biological Services under Protocol No. PPL/70/7900. Mice were generated to overexpress c-Jun selectively in Schwann cells as described (*Fazal et al., 2017*). Briefly, female *R26c-Junstopf* mice carrying a lox-P flanked STOP cassette in front of a CAG promoter-driven c-Jun cDNA in the ROSA26 locus, were crossed with male *Mpz^Cre+/−* mice (*Feltri et al., 1999*). This generated *Mpz^Cre+;R26c-Junstop^ff/+* mice, referred to here as c-Jun OE/+ mice. *Mpz^Cre−/Cre−* littermates were used as controls. *Shh-floxed* mice, referred to as Shh cKO mice, carrying *loxP* sites flanking exon 2 of the *Shh* gene were obtained from the Jackson Laboratory (Jax, stock# 004293), and bred to *Mpz^Cre* mice (*Feltri et al., 1999*). Experiments used mice of either sex on the C57BL/6 background.

## Genotyping

DNA was extracted from ear notches or tail tips using the Hot Sodium Hydroxide and Tris method (HotSHot; *Truett et al., 2000*). Tissue was incubated in HotSHot buffer (25 mM NaOH and 0.2 mM disodium EDTA, pH 12) at 95℃ for 1 hr. The reaction was neutralized with neutralizing buffer (40 mM Tris-HCl, pH 5). DNA was then added to the PCR mastermix with primers for the *Mpz-Cre* transgene: 5′-GCTGGCCCAAATGTTGCTGG-3′ and 5′CCACCACCTCTCCATTGCAC-3′ (480 bp band).

## Surgery

For short-term time points (<1 week) and crushes, the sciatic nerve was exposed and cut or crushed (at three rotation angles) at the sciatic notch. For western blot, immunofluorescence and mRNA

investigations into chronic denervation (>1 week) the sciatic nerve was cut and the proximal stump was reflected back and sutured into muscle to prevent regeneration. The nerve distal to the injury was excised for analysis at various time points. Contralateral uninjured sciatic nerves served as controls. To examine the effects of chronic denervation on regeneration, the nerve branches of the sciatic nerve were individually separated (*Figure 2—figure supplement 1*). The tibial nerve was cut and both proximal and distal stumps were reflected and sutured into muscle. Either immediately or following 10 weeks of chronic denervation, the distal tibial nerve stump was cut from the muscle and sutured to the freshly transected common peroneal nerve.

## Retrograde labeling with Fluorogold

To examine regeneration following nerve crush or repair, the nerve was cut distal to the original injury site and exposed to 4% Fluorogold for 1 hr (*Catapano et al., 2016*; *Figure 1—figure supplement 1*). The spinal cord and L4 DRG were removed following perfusion 1 week post-labeling. Labeled cells in all the spinal cord sections (50 μm) were counted and the Abercrombie correction was applied to compensate for double counting (*Abercrombie, 1946*). To avoid double counting, cells in every fifth DRG section (20 μm) were counted.

## Schwann cell cultures

Rat Schwann cells were cultured as described (*Brockes et al., 1979*). Briefly, sciatic nerves and brachial plexuses were digested enzymatically with collagenase and trypsin and cultured on laminin- and PLL-coated plates in DMEM, 2% FBS, 10 ng/ml NRG-1, 2 μM forskolin and penicillin/streptomycin. Under experimental conditions, cultures were maintained in defined medium (DMEM and Ham's F12 (1:1), transferrin (100 pg/ml), progesterone (60 ng/ml), putrescine (16 pg/ml), insulin (5 μg/ml), thyroxine (0.4 μg/ml), selenium (160 ng/ml), triiodothyronine (10.1 ng/ml), dexamethasone (38 ng/ml), glucose (7.9 mg/ml), bovine serum albumin (0.3 mg/ml), penicillin (100 IU/ml), streptomycin (100 IU/ml), and glutamine (2 mM) with 0.5% serum *Jessen et al., 1994*; *Meier et al., 1999*).

## Antibodies

Immunofluorescence antibodies: c-Jun (Cell Signaling Technology, rabbit 1:800), Sox10 (R and D Systems, goat 1:100), CGRP (Peninsula, rabbit 1:1000), neurofilament (Abcam, rabbit 1:1000), donkey anti-goat IgG (H+L) Alexa Fluor 488 conjugate (Invitrogen, 1:1000), and Cy3 donkey anti-rabbit IgG (H+L) (Jackson Immunoresearch, 1:500).

Antibodies used for western blotting: c-Jun (Cell Signaling Technology, rabbit 1:1000), p75 NTR (Millipore, rabbit 1:1000), serine 63 phosphorylated c-Jun (Cell Signaling Technology, rabbit 1:1000), GAPDH (Sigma-Aldrich, rabbit 1:5000), calnexin (Enzo Life Sciences, rabbit 1:1000), and anti-rabbit IgG, HRP-linked (Cell Signaling Technology, 1:2000).

## Immunofluorescence

For immunofluorescence experiments on cultured cells, 5000 Schwann cells were plated in a 35 μl drop on a PLL laminin-coated coverslip. Cells were topped up with defined medium after 24 hr. At the experimental end point, cells were washed 2x with 1x PBS. Cells were fixed with 4% paraformaldehyde ( PFA) for 10 min. Cells were then washed for 5 min in 1x PBS. Fresh PBS was added to the wells and the lid was parafilm sealed. Dishes were stored at 4°C until use.

Nerve samples were fresh frozen during embedding in OCT. Cryosections were cut at 10 μm. Sections were fixed in 100% acetone (Sox10/c-Jun double-labeling, 10 min at −20°C) or 4% PFA (10 min at room temperature).

For immunofluorescence, all samples were washed 3x in 1x PBS and blocked in 5% donkey serum, 1% BSA, 0.3% Triton X-100 in PBS. Samples were incubated with primary antibodies in blocking solution overnight at 4°C. Sox10/c-Jun double-labeling was performed overnight at room temperature. Samples were washed and incubated with secondary antibodies and DAPI to identify cell nuclei (Thermo Fisher Scientific, 1:40,000) in PBS for 1 hr at room temperature. Samples were mounted in fluorescent mounting medium (Citifluor).

Images were taken on a Nikon Labophot two fluorescence microscope. Cell counts were performed in ImageJ or directly from the microscope. Comparable images have been equally adjusted

for brightness/contrast. In some cases (*Figures 1*, *3* and *4*), images of whole nerve profiles have been generated by stitching together multiple images.

## Western blotting

Nerves were dissected and snap frozen in liquid nitrogen. For protein extraction, nerves were placed in 2 ml graduated skirted tubes with nine 10B lysing beads with 75 ml lysis buffer (1M Tris-HCl pH 8, 5M NaCl, 20% Triton X-100, 5 mM EDTA) and homogenized using a Fastprep fp120 homogeniser. Samples were run twice at speed 6 for 45 s. Lysates were then centrifuged at 13,000 rpm for 2 min at 4°C to pellet the debris. The supernatant was transferred to a new 1.5 ml Eppendorf tube and centrifuged at 13,000 rpm for 2 min at 4°C. The supernatant was transferred to a new 1.5 ml Eppendorf tube and the protein extract was stored at −80°C.

For protein studies on cultured cells, $1 \times 10^6$ purified Schwann cells were plated in a 35 mm dish in defined medium for 48 hr. At the time of extraction, the cultures were washed 2x with 1x PBS and incubated with 100 µl cell lysis buffer (T-PER Tissue Protein Extraction Reagent, Halt protease, and phosphatase inhibitor cocktail [1:100] Thermo Fisher Scientific). Cells were physically detached from dishes using a cell scraper. The cell lysate was collected and kept on ice in a 1.5 ml Eppendorf tube. Lysate was spun for 2 min at 1000 rpm to pellet the debris. The supernatant was transferred to a fresh Eppendorf tube and spun for a further 2 min at 1000 rpm. The supernatant was transferred to a new 1.5 ml Eppendorf tube and stored at −80°C until use.

Protein was diluted in the appropriate lysis buffer and 5x Laemmli buffer at a working concentration of 1x. Samples were heated to 95°C for 5 min to denature the protein. 10 µg protein was loaded per well on 8% acrylamide gels with prestained standard molecular weight markers (PageRuler prestained protein ladder; Thermo Fisher Scientific) and run at 60 mV for 3 hr using the mini Protean II gel electrophoresis apparatus (Bio-Rad Laboratories). Protein was transferred to a nitrocellulose membrane (Hybond ECL; Amersham Biosciences) using a semi-dry transfer system (Bio-Rad Laboratories) at 25 mV for 45 min. Membranes were briefly stained with Ponceau S (Sigma Aldrich) to determine that the transfer has been successful and that equal levels of protein had been loaded in the gel. Membranes were briefly washed in ddH$_2$O to remove excess Ponceau and blocked in 5% milk/TBS-T for 1 hr with shaking at room temperature. Membranes were then incubated with appropriate antibodies in heat sealable polyethylene bags and were incubated overnight at 4°C on a rotatory wheel. Membranes were washed 3x for 10 min in 1x TBS-T then incubated with the appropriate secondary antibody in heat sealable polyethylene bags, rotating for 1 hr at room temperature. Membranes were washed 3x for 10 min in 1x TBS-T before developing. For development of GAPDH, membranes were incubated with ECL (Amersham) for 1 min and developed on a Bio-Rad Chemidoc machine. For the development of all other antibodies, membranes were incubated with ECL prime (Amersham) for 5 min then developed. Membranes were automatically exposed to prevent saturation. Blots were analyzed and densitometric quantification was performed using Bio-Rad Imagelab. Protein levels were determined by normalizing the protein of interest against the house keeping protein (GAPDH or calnexin). All blots were then normalized to one sample (e.g. 1 week after injury, control cells) to account for any difference between each blot. Each experiment was performed at least three times with fresh samples. Representative images are shown.

## Electron microscopy

Nerves were fixed in 2.5% glutaraldehyde/2% paraformaldehyde in 0.1 M cacodylate buffer, pH 7.4, overnight at 4°C. Post-fixation in 1% OsO$_4$ was performed before nerves were embedded in Agar 100 epoxy resin. Transverse ultrathin sections from adult (P60) or aged (P300) tibial nerves or from injured distal stumps of adult sciatic nerves at various times after injury were taken 5 mm from the sciatic notch and mounted on film (no grid bars). Images were examined using a Jeol 1010 electron microscope with a Gatan camera and software. Images were examined and photographed at 8000 $\times$ or 15,000x. The nerve area was measured from photographs taken at 200 $\times$ magnification. Schwann cells and macrophages and fibroblasts were identified by standard ultrastructural criteria (e.g. *Reichert et al., 1994*). Schwann cell, macrophage, and fibroblast nuclei were counted in every field, or every second, third, or fourth field, depending on the size of the nerve, and multiplied by the number of fields to generate totals.

## qPCR

RNA from rat Schwann cell cultures or mouse nerve tissue was extracted using an RNeasy Micro Extraction Kit (Qiagen). RNA quality and concentration was determined after extraction using a nanodrop 2000 machine (Thermo). One μg of RNA was converted to cDNA using SuperScriptTM II Reverse Transcriptase (Invitrogen) as per the manufacturer's instructions. Samples were run with Pre-cisionPLUS qPCR Mastermix with SYBR Green (Primerdesign) with primers as described in *Benito et al., 2017*. Ct values were normalized to housekeeping gene expression (*GAPDH and calnexin*).

| Primer | Sequence 5´- 3´ |
|---|---|
| *Bdnf* Fwd | TCATACTTCGGTTGCATGAAGG |
| *Bdnf* Rev | AGACCTCTCGAACCTGCCC |
| *c-Jun* Fwd (Cells) | AATGGGCACATCACCACTACAC |
| *c-Jun* Rev (Cells) | TGCTCGTCGGTCACGTTCT |
| *c-Jun* Fwd (Tissue) | CCTTCTACGACGATGCCCTC |
| *c-Jun* Rev (Tissue) | GGTTCAAGGTCATGCTCTGTTT |
| *Gdnf* Fwd | GATTCGGGCCACTTGGAGTT |
| *Gdnf* Rev | GACAGCCACGACATCCCATA |
| *Calnexin* Fwd | CAACAGGGGAGGTTTATTTTGCT |
| *Calnexin* Rev | TCCCACTTTCCATCATATTTGGC |
| *GAPDH* Fwd | AGGTCGGTGTGAACGGATTTG |
| *GAPDH* Rev | TGTAGACCATGTAGTTGAGGTCA |

## Statistical analysis

Results are expressed as mean ± SEM. Statistical significance was estimated by Student's *t* test, one-way ANOVA or two-way ANOVA with appropriate post hoc tests. A p value < 0.05 was considered as statistically significant. Statistical analysis was performed using GraphPad software.

## Library preparation

RNA was extracted using a RNeasy lipid tissue kit with an in column DNase step (Qiagen). Chronically denervated and uninjured nerves were pooled, two per n. Acutely denervated nerves were not pooled using one nerve per n. The library was prepared using the Kapa mRNA Hyper Prep kit (Roche) with 100 ng RNA and 15 cycles of PCR enrichment. The assay is (first) stranded (dUTP method).

## Sequencing

Sequencing was performed in a pooled NextSeq 500 run using a 43 bp paired end protocol (plus a 6 bp index read). Sequencing reads (in fastq format) were aligned to the hg38 reference sequence using STAR v2.5.3 (*Dobin et al., 2013*). Samtools version 1.2 and Picard tools version 1.140 were used to process alignments (*Li et al., 2009*), Aligned reads were filtered for mapq _ 4 that is uniquely mapping reads, and putative PCR duplicates were removed. featureCounts was used to perform read summarization (*Liao et al., 2014*).

## Data analysis

Expression analysis was carried out using R version 3.5.1. Differential gene expression was analyzed using edgeR (*Robinson et al., 2010*). Genes with both an absolute log2 fold change >2.0 and FDR < 0.05 were identified as being significantly differentially expressed. Principal component analysis (PCA) showed that injury status was the dominant source of variation in both data sets (*Figure 9—figure supplement 1A*; *Figure 10—figure supplement 1A*). Enrichment of c-Jun-regulated genes was investigated using Fisher one-sided exact tests and GSEA ( *Subramanian et al., 2005*). GO

analysis was used to examine gene enrichment of all significantly differentiated genes using the PANTHER classification system (*Mi et al., 2013*).

## Additional information

### Funding

| Funder | Grant reference number | Author |
|---|---|---|
| EC Seventh Framework Programm | HEALTH F2 2008-201535 | Laura J Wagstaff |
| Wellcome Trust | Programme Grant 074665 | Kristjan R Jessen |
| Wellcome Trust | PhD studentship 511851 FBAGG F96 WT Stern (No 161 761 | Rhona Mirsky |
| Medical Research Council | Project grant G0600967) | Rhona Mirsky Kristjan R Jessen |
| Agencia Estatal de Investigación | BFU2016-75864R | Hugo Cabedo |
| Agencia Estatal de Investigación | PID2019-109762RB-I00 | Hugo Cabedo |
| Conselleria de Sanitat, Generalitat Valenciana | PROMETEO 2018/114 | Hugo Cabedo |
| National Institute of Child Health and Human Development | HD090256 | John Svaren |
| National Institute of Neurological Disorders and Stroke | NS075269 | John Svaren |
| National Institute of Neurological Disorders and Stroke | NS100510 | John Svaren |
| Wellcome Trust | 206634 | Peter Arthur-Farraj |

The funders had no role in study design, data collection and interpretation, or the decision to submit the work for publication.

### Author contributions

Laura J Wagstaff, Shaline V Fazal, Data curation, analysis, validation,investigation, visualization, methodology, writing.; Jose A Gomez-Sanchez, Data curation, Supervision, Investigation, Visualization, Methodology; Georg W Otto, Data curation, Formal analysis; Alastair M Kilpatrick, Data curation. Formal analysis,; Kirolos Michael, Liam YN Wong, Ki H Ma, Mark Turmaine, Sergio Velasco-Aviles, Cristina Benito, Investigation; John Svaren, Hugo Cabedo, Resources, Investigation; Tessa Gordon, Supervision, Investigation; Peter Arthur-Farraj, Formal analysis; Rhona Mirsky, Conceptualization, Supervision, Investigation, Writing - original draft, Project administration, Writing - review and editing; Kristjan R Jessen, Conceptualization, Funding acquisition, Investigation, Writing - original draft, Project administration, Writing - review and editing

### Author ORCIDs

Jose A Gomez-Sanchez ORCID https://orcid.org/0000-0002-6746-1800
Georg W Otto ORCID https://orcid.org/0000-0002-3929-948X
Alastair M Kilpatrick ORCID https://orcid.org/0000-0002-4795-8799
Kirolos Michael ORCID https://orcid.org/0000-0002-2547-9869
Ki H Ma ORCID http://orcid.org/0000-0003-3474-7287
John Svaren ORCID http://orcid.org/0000-0003-2963-7921
Peter Arthur-Farraj ORCID https://orcid.org/0000-0002-1239-9392
Kristjan R Jessen ORCID https://orcid.org/0000-0002-7883-1221

## Ethics

Animal experimentation: Animal experiments conformed to UK Home Office guidelines under the supervision of University College London (UCL) Biological Services under Protocol No. PPL/79/7900.

## Decision letter and Author response

Decision letter https://doi.org/10.7554/eLife.62232.sa1
Author response https://doi.org/10.7554/eLife.62232.sa2

# Additional files

### Supplementary files

• Supplementary file 1. The 15 most regulated genes in the tibial nerve of WT mice during aging. The data compare (A) uncut nerves, (B) 3-day cut nerves, and (C) the injury response. In (A) and (B), genes expressed at higher levels in aged mice under the conditions indicated are in blue (top), while genes with reduced expression in aged mice are in red (bottom). In (C), genes that respond more strongly to injury in aged mice are in blue (top), while genes with weaker injury response in aged mice are in red (bottom).

• Supplementary file 2. All significantly regulated genes in the tibial nerve of WT mice during aging and chronic denervation. (A, B) Genes expressed at higher levels in aged mice under the conditions indicated are in blue (top), while genes with reduced expression in aged mice are in red (bottom). (C) Genes that respond more strongly to injury (3-day cut vs UI) in aged mice are in blue (top), while genes with weaker injury response are in red (bottom). (D) The 303 genes that are significantly expressed in young WT vs UI and aged c-Jun OE/+ vs UI 3 days after injury. (E) Genes expressed at higher levels after chronic denervation are in blue (top), while genes with reduced expression are in red (bottom). (F) The 227 genes that are significantly expressed in WT 1 week vs UI and c-Jun OE/+ 10 week vs UI.

• Supplementary file 3. 138 genes regulated by c-Jun in injured nerves derived from *Arthur-Farraj et al., 2012*. Blue (top) indicates genes expressed at higher levels in cut nerves of WT mice compared with nerves of mice with conditional c-Jun inactivation selectively in Schwann cells (92 genes), while red (bottom) indicates genes expressed at lower levels in cut WT nerves compared to c-Jun mutant nerves (46 genes).

• Supplementary file 4. The 15 most regulated genes in the tibial nerve during chronic denervation. Genes expressed at higher levels after chronic denervation are in blue (top), while genes with reduced expression are in red (bottom).

• Transparent reporting form

### Data availability

Sequencing data have been submitted to Arrayexpress.

The following dataset was generated:

| Author(s) | Year | Dataset title | Dataset URL | Database and Identifier |
|---|---|---|---|---|
| Otto G, Jessen KR, Mirsky R, Wagstaff LJ, Gomez-Sanchez JR | 2017 | Failures of nerve regeneration caused by aging and chronic denervation are rescued by restoring Schwann cell c-Junn | https://www.ebi.ac.uk/arrayexpress/experiments/E-MTAB-9640/ | ArrayExpress, E-MTAB-9640 |

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
