## [Decision Letter]

**Acceptance summary:**

This study demonstrates that aging and chronic injury of peripheral nerves can be overcome by Schwann cell signaling. Regeneration is mediated through upregulation of the c-Jun factor. High levels of Schwann cell c-Jun is a key event in neuronal regeneration. Restoring Schwann cell c-Jun levels restores regeneration.

**Decision letter after peer review:**

Thank you for submitting your article "Failures of nerve regeneration caused by aging or chronic denervation are rescued by restoring Schwann cell c-Jun" for consideration by *eLife*. Your article has been reviewed by three peer reviewers, and the evaluation has been overseen by a Reviewing Editor and Marianne Bronner as the Senior Editor. The following individual involved in review of your submission has agreed to reveal their identity: Ahmet Hoke (Reviewer #1).

The reviewers have discussed the reviews with one another and the Reviewing Editor has drafted this decision to help you prepare a revised submission.

Summary:

This manuscript demonstrates that upregulation of c-Jun in aged and chronically denervated nerves is sufficient to restore reduced regenerative support by Schwann cells. A reduction of c-Jun in both aged Schwann cells and with chronic denervation was known. The hypothesis that upregulation of c-Jun would enhance regenerative success is amply documented by this study. The authors also report that c-Jun is controlled by Shh, which is a novel finding. This paper is well done and adds an important contribution to the literature.

A strength of this study is the ability to compare two different injury models. Additionally, many experiments are detailed and done in a rigorous fashion.

Although the models used are appropriate, I have some concerns about them. For example, in the aging model, I would have preferred to see examination of success of regeneration at a time point later than 4 days and at a distance longer than just 7 mm. Given short distance from crush site, there is always a risk that retrograde labeling can extend proximally and in fact label some of the axons that may not even have been injured.

Also, it is a bit odd that they have equal numbers of labeled neurons in DRG and spinal cord when in fact 80% of axons in a sciatic nerve as sensory axons (Figure 1G and I).

The authors need to be commended for developing the mouse chronic denervation model, but, similarly, retrograde labeling at only 4 mm distal to cut and repair lends itself to technical challenges and possibility of label diffusing into the more proximal segments of the nerve.

---

## [Author Response]

Summary:This manuscript demonstrates that upregulation of c-Jun in aged and chronically denervated nerves is sufficient to restore reduced regenerative support by Schwann cells. A reduction of c-Jun in both aged Schwann cells and with chronic denervation was known.

As far as we are aware, the down-regulation of c-*Jun D*uring chronic denervation was not known previously.

The hypothesis that upregulation of c-Jun would enhance regenerative success is amply documented by this study. The authors also report that c-Jun is controlled by Shh, which is a novel finding. This paper is well done and adds an important contribution to the literature.A strength of this study is the ability to compare two different injury models. Additionally, many experiments are detailed and done in a rigorous fashion.Although the models used are appropriate, I have some concerns about them. For example, in the aging model, I would have preferred to see examination of success of regeneration at a time point later than 4 days and at a distance longer than just 7 mm. Given short distance from crush site, there is always a risk that retrograde labeling can extend proximally and in fact label some of the axons that may not even have been injured.

The small size of the mouse leg, compared to the rat, sets an upper limit to the distance between the crush site and regeneration distance (i.e. the site of the second injury, the point at which the retrograde label is applied). See also discussion on this point below regarding chronic denervation. The consistency of the results, and in particular the consistent differences between WT and c-Jun OE/+ mice, strongly argues against accidental labelling in some experiments of axons that had not reached the reference-distance (7mm). If the retrograde label had reached for significant distances in the proximal direction, we would not expect to see a difference between OE/+ and WT mice as each have equal numbers of sensory and motor neurons that become labelled in this type of experiment (Figure 5F and G).

Also, it is a bit odd that they have equal numbers of labeled neurons in DRG and spinal cord when in fact 80% of axons in a sciatic nerve as sensory axons (Figure 1G and I).

Only neurons in L4 were analysed in these experiments. This explains the relatively low number of labelled DRG neurons, relative to the number of sensory axons in the sciatic nerve.

The authors need to be commended for developing the mouse chronic denervation model, but, similarly, retrograde labeling at only 4 mm distal to cut and repair lends itself to technical challenges and possibility of label diffusing into the more proximal segments of the nerve.

Establishing the chronic denervation model in the mouse was technically challenging due to the small size of mouse nerves. As we worked on this with one of the leaders in the field, Tessa Gordon, and in her laboratory, we found that 4mm was the furthest distance that could be examined due to space limitations. Our protocol is based on previous work by Tessa Gordon who pioneered this type of operation., and is a co-author of the present manuscript.

In these experiments, there is no question of retrograde label reaching uninjured axons, since the experiment involves cutting (not crushing) the nerves in question.